# Spatial readout of visual looming in the central brain of *Drosophila*

**Mai M Morimoto[1,2], Aljoscha Nern[1†], Arthur Zhao[1†], Edward M Rogers[1], Allan M Wong[1], Mathew D Isaacson[1,3], Davi D Bock[1,4], Gerald M Rubin[1], Michael B Reiser[1]***

[1]Janelia Research Campus, Howard Hughes Medical Institute, Ashburn, United States; [2]Department of Experimental Psychology, University College London, London, United Kingdom; [3]Department of Biomedical Engineering, Cornell University, Ithaca, United States; [4]Department of Neurological Sciences, University of Vermont, Burlington, United States

**Abstract** Visual systems can exploit spatial correlations in the visual scene by using retinotopy, the organizing principle by which neighboring cells encode neighboring spatial locations. However, retinotopy is often lost, such as when visual pathways are integrated with other sensory modalities. How is spatial information processed outside of strictly visual brain areas? Here, we focused on visual looming responsive LC6 cells in *Drosophila*, a population whose dendrites collectively cover the visual field, but whose axons form a single glomerulus—a structure without obvious retinotopic organization—in the central brain. We identified multiple cell types downstream of LC6 in the glomerulus and found that they more strongly respond to looming in different portions of the visual field, unexpectedly preserving spatial information. Through EM reconstruction of all LC6 synaptic inputs to the glomerulus, we found that LC6 and downstream cell types form circuits within the glomerulus that enable spatial readout of visual features and contralateral suppression—mechanisms that transform visual information for behavioral control.

**\*For correspondence:**
reiserm@janelia.hhmi.org

[†]These authors contributed equally to this work

**Competing interests:** The authors declare that no competing interests exist.

## Introduction

In many animals, brain regions involved in processing visual information are large and well organized, featuring retinotopy—an organizational plan that preserves the mapping of space originating in the retina, such that neighboring neurons respond to visual signals at neighboring spatial locations. Animals need visual-spatial information in order to direct their escape away from predators, to find mates, or to catch prey. Classical studies in cats (*Hubel and Wiesel, 1962*) and monkeys (*Tootell et al., 1988*), and recent work in mice (*Garrett et al., 2014*) demonstrated that the retinotopic organization of higher visual areas is a dominant feature of the organization of the mammalian brain. Although this organizing principle has facilitated detailed analyses of visual pathways, it remains unclear how visual-spatial information is processed outside of strictly visual brain regions. In particular, retinotopy is likely sacrificed where visual information is integrated with other modalities and as vision is translated into behavioral actions. We propose that rapid progress can be made on understanding these critical transformations in the *Drosophila* brain.

The lobula columnar (LC) neurons project from the visual system into the central brain of flies and have been well described in *Drosophila* (*Fischbach and Dittrich, 1989*; *Otsuna and Ito, 2006*; *Wu et al., 2016*). They are positioned at the last processing step of a retinotopic neuropil and are a nexus between the detection of visual features and the organization of behavioral control. There are ~20 types of LC cells, and we have previously shown that optogenetic activation of individual cell types elicits a wide range of behaviors that closely resemble natural behaviors such as escape jumps, backward walking, courtship behavior, and reaching (*Wu et al., 2016*). Furthermore, the cell types

that elicited avoidance behaviors (LC4, LPLC2, LC6, and LC16) also responded to visual looming stimuli (*Ache et al., 2019*; *Klapoetke et al., 2017*; *Sen et al., 2017*; *Wu et al., 2016*). Other cell types, including cells whose activation elicited courtship or reaching behaviors (LC10), responded to the motion of small visual objects (*Keleş and Frye, 2017*; *Ribeiro et al., 2018*; *Wu et al., 2016*). These findings suggest that LC neurons encode ethologically relevant visual object features and that these are translated into appropriate visually guided actions by downstream circuits. Anatomical studies, based on light microscopy, have suggested that most LC neurons (with the notable exception of LC10) lose retinotopy—their inputs in the visual system are clearly organized with the spatial layout of the retina, while their axonal projections, into glomeruli in the central brain, appear to lack any continuous visual-spatial organization (*Mu et al., 2012*; *Wu et al., 2016*). This anatomical evidence has led to the suggestion that LC neurons may represent a critical transition from visual ('where') processing to a more abstract ('what') representation of visual information (*Mu et al., 2012*; *Strausfeld et al., 2007*; *Wu et al., 2016*).

There is some evidence for retinotopic organization in specific regions of the fly central brain. While most LC glomeruli do not exhibit retinotopic organization that can be detected by light microscopy analysis, the LC10 projection to the anterior optic tubercle (AOTu) is a notable exception (*Wu et al., 2016*). LC10s are known to mediate visual components of courtship behavior and are therefore thought to convey the spatial position of conspecifics in the visual field (*Ribeiro et al., 2018*). Retinotopy was also observed in a different visual pathway, via another part of the AOTu. Functional studies revealed that the ellipsoid body ring neurons exhibit clear visual-spatial organization with some retinotopy; these neurons supply the visual signals that are used to update the fly's heading system (*Fisher et al., 2019*; *Kim et al., 2019*; *Seelig and Jayaraman, 2013*). The anatomical basis of this functional retinotopy is the projection from the medulla to the bulb, via the optic tubercle, which is known to feature a strong degree of retinotopic organization (*Omoto et al., 2017*; *Sun et al., 2017*; *Timaeus et al., 2020*). Beyond these examples, the extent to which central projections from the *Drosophila* optic lobe convey retinotopic organization, and thus participate in spatial vision, is not well known.

In this study, we have systematically explored the downstream circuitry of LC6 (*Figure 1A–B,B'*), a cell type that elicited the most reliable escape take-off behavior when depolarized (optogenetic activation screen; *Wu et al., 2016*), but is not a direct input to the well-studied giant-fiber escape pathway (*Ache et al., 2019*). The LC6 axons are presynaptic within the LC6 glomerulus, where no retinotopic organization can be discerned using light microscopy (*Figure 1C,C', D*). We identified several neurons with arbors in the LC6 glomerulus and established genetic tools for targeting these cell types. We assayed the functional connectivity between LC6 neurons and these candidate downstream cell types, yielding five distinct cell types that integrate LC6 inputs. We further investigated two of these downstream neuron types, one that connected the two LC6 glomeruli across both hemispheres, and an ipsilateral type that projected to a higher order multi-sensory area. Using calcium imaging, we examined whether these downstream targets relayed or transformed the signals encoded by their LC6 inputs. During this analysis, we were surprised to find that LC6 target neurons read out spatially biased information from the LC6 glomerulus, despite the lack of prominent retinotopic organization in this structure.

What if visual-spatial information is conveyed not by the innervation pattern of these projections, but rather by specific synaptic connections? To determine where this specificity for spatial information originates, we undertook a comprehensive analysis of the connectivity between LC6 neurons and two downstream cell types using whole-brain Electron Microscopy (EM) data (*Zheng et al., 2018*). We reconstructed 'anatomical' receptive fields for all LC6 neurons and used them to estimate the receptive fields of the connected downstream cells. We found that downstream neurons access distinct spatial information via biased connections with LC6s, consistent with our functional measurements. Finally, combining connectivity and functional data, we detail a circuit wherein a downstream target of LC6 receives contralateral suppression that could enhance the detection of looming stimuli over confounding visual cues.

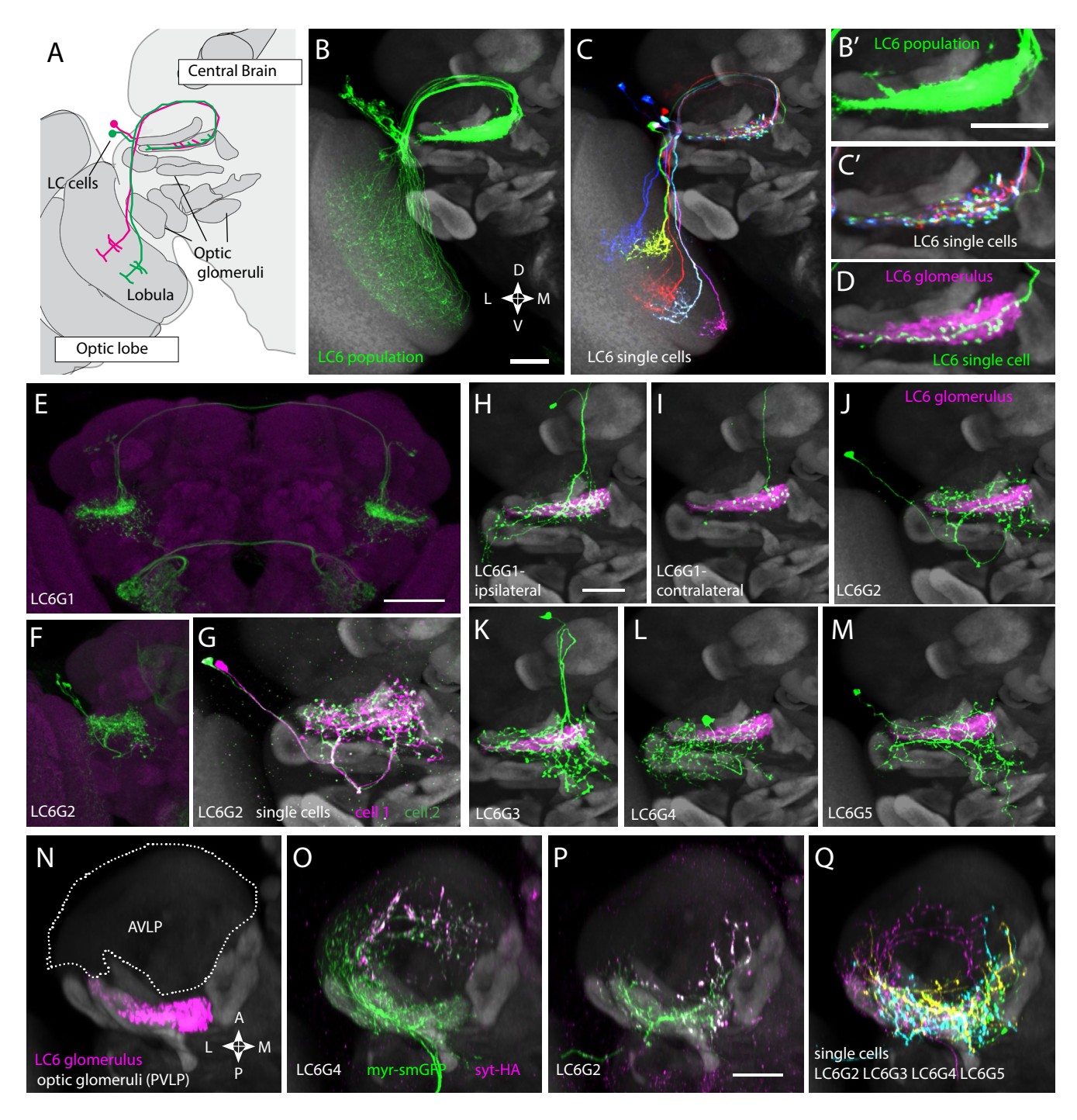

**Figure 1.** Multiple distinct cell types innervate the LC6 glomerulus. (A–D) Projection pattern of LC6 neurons and (E–Q) anatomy of candidate LC6 target cells. All panels (except E), (F) show composites of registered confocal images together with the standard brain used for registration (shown in gray). To more clearly show the neurons of interest, some images were manually segmented to exclude additional labeled cells or background signal (see Materials and methods). (A) Lobula Columnar (LC) neurons project from the lobula to synapse-rich structures in the posterior ventrolateral protocerebrum (PVLP) called optic glomeruli. Two LC6 cells projecting to the LC6 glomerulus are illustrated. Schematic adapted from *Wu et al., 2016*. In all panels except for N-Q dorsal is up and medial is to the right. (B, C) LC6 neuron dendrites collectively tile the lobula, while the axons converge to form a glomerulus. (B, B') Population of LC6 neurons labeled with a membrane targeted marker. Scale bar, 20 μm. (C, C') MultiColor FlpOut (MCFO)-labeled individual LC6 cells registered and displayed as in (B). Note that individual LC6 terminals overlap in the LC6 glomerulus, while dendrites in the lobula occupy distinct positions. (D) Terminal of a single MCFO-labeled LC6 cell displayed as in (B') with the LC6 glomerulus, based on the expression

*Figure 1 continued on next page*

Figure 1 continued

of syt-HA, a tagged synaptotagmin in LC6 cells as described in *Wu et al., 2016*, in magenta. (E, F) Examples of two populations of candidate LC6 target cells labeled by split-GAL4 driver lines. A membrane marker is in green, a synaptic marker (anti-Brp) in magenta. Scale bar, 50 μm. (G) MCFO labeling of two LC6G2 cells in the same specimen. (H–M) Examples of single cells of potential LC6 targets. All images show MCFO-labeled manually segmented single cells (green) together with the standard brain (gray) and LC6 glomerulus (magenta). Scale bar, 20 μm. (N–O) Several LC6 glomerulus interneurons project to the anterior ventrolateral protocerebrum (AVLP). The images show projections through a sub stack rotated around the mediolateral axis by 90° relative to the view shown in panels (A–M) with the reference brain neuropil marker (gray). (N) LC6 glomerulus (magenta) and approximate boundaries of the AVLP (dotted white line). (O, P) LC6G4 and LC6G2 populations, as labeled by split-GAL4 lines. In addition to the membrane label (green) and standard brain (gray) a presynaptic marker (syt-HA) is shown (magenta). Note syt-HA labeling in the AVLP. (Q) Overlay of the segmented single cells of LC6G2, LC6G3, LC6G4, and LC6G5 shown in (J–M) with the standard brain. Images in (N–Q) are sub stack projection, some cells in (O–Q) have additional AVLP branches outside of the projected volume. See *Supplementary file 1* for detailed summary of fly lines used in this study.

The online version of this article includes the following figure supplement(s) for figure 1:

**Figure supplement 1.** Additional anatomical details of LC6G neurons.
**Figure supplement 2.** Expression patterns of LC6G split-GAL4 lines.
**Figure supplement 3.** FISH detection of neurotransmitter markers in LC6G neurons.

## Results

### Multiple distinct cell types innervate the LC6 glomerulus

In *Drosophila*, it has become increasingly feasible to identify genetic markers for cell types of interest by analyzing images of GAL4 line expression patterns (*Jenett et al., 2012*; *Kvon et al., 2014*; *Panser et al., 2016*). To find potential LC6 targets, we visually searched for neurons with processes that substantially overlap with the LC6 glomerulus, a distinct neuropil structure that contains the densely packed terminals of LC6 neurons (*Figure 1A–D*). This approach identified several candidate LC6 glomerulus interneurons (*Figure 1*; *Figure 1—figure supplement 1*). To facilitate studies of these cells, we developed more specific split-GAL4 driver lines (*Luan et al., 2006*; *Pfeiffer et al., 2010*; *Figure 1E,F*, *Figure 1—figure supplement 2*). We obtained split-GAL4 lines with expression in each of five distinct cell populations that heavily overlap with the LC6 glomerulus. We named these putative cell types, LC6G1 - LC6G5 (short for LC6 Glomerulus 1–5; *Figure 1H–M*, *Figure 1—figure supplement 1*). LC6G types have identifying characteristic cell body locations and projection patterns (*Figure 1E–F*, *Figure 1—figure supplements 1* and *2*). Cell body positions, presumably reflecting distinct developmental origins, are in the cell body rind of the anterior dorsal (LC6G1 and LC6G3) or posterior (LC6G4) central brain and in the space between the optic lobe and central brain (LC6G2 and LC6G5). Bilateral arbors that span both brain hemispheres differentiate LC6G1 from the other LC6Gs (which have only ipsilateral processes; *Figure 1E,F*, *Figure 1—figure supplement 2*).

Each LC6G split-GAL4 driver labels multiple individual cells (disregarding unrelated cells in distinct regions also present in some lines). For several driver lines, we used multicolor stochastic labeling (*Nern et al., 2015*) to directly visualize multiple similar cells in the same specimen (*Figure 1G*, *Figure 1—figure supplement 1*). There are about 4–6 LC6G1, LC6G2, LC6G3, and LC6G4 cells each per hemisphere. While these numbers could include a few unrecognized cells of other types or underestimate the true cell number due to incomplete driver expression, each LC6G type appears to consist of multiple cells of very similar or identical morphology. Similar to LC6 cells (*Figure 1C,D*), the distribution of processes of LC6G cells does not subdivide the LC6 glomerulus into distinct subregions. If these neurons are indeed LC6 targets, their anatomy does not indicate whether they receive input from most or all LC6 cells or from random or specific subsets.

Expression of neurotransmitter markers (VGlut, GAD1, ChaT) also indicates differences between LC6G types: LC6G1 and LC6G3 appear to be glutamatergic, LC6G4 GABAergic, and LC6G2 and LC6G5 cholinergic (*Figure 1—figure supplement 3*). In the fly CNS, while glutamate is known to act as both an excitatory or an inhibitory transmitter, widespread expression of a glutamate-gated chloride channel (including in LC6—*Davis et al., 2020*) indicates that glutamate frequently acts as an inhibitory neurotransmitter (*Liu and Wilson, 2013*; *Mauss et al., 2015*; *Strother et al., 2017*), suggesting that the LC6G1, LC6G3, and LC6G4 neurons may provide inhibition of their targets in the LC6 glomerulus. On the other hand, LC6G2, LC6G5, and LC6 neurons express ChaT (*Figure 1—figure supplement 3* and *Davis et al., 2020*) and so are likely to be cholinergic, and thus excitatory

neurons. Single-cell labeling revealed further details of each group (*Figure 1H–M,G*, *Figure 1—figure supplement 1*). While we focus here on the LC6 glomerulus, our search also identified neurons with arbors in other glomeruli. One example, a potential LC26 target (LC26G1) is shown in *Figure 1—figure supplement 1*; we use this cell type as a control in *Figure 2—figure supplement 1*.

All LC6G cells have at least some branches outside of the glomerulus. In several cases, these extend into other glomeruli, including, for example, the LC16 target region in the case of LCG6G2 (*Figure 1J*) and the LC15 and LC21 glomeruli for LC6G4 (*Figure 1L*). Apart from other glomeruli and their immediate surroundings, the AVLP (Anterior Ventrolateral Protocerebrum) appears to be the main target region for ipsilateral LC6G cells (*Figure 1N–Q*). In particular, LC6G2, LC6G4, and LC6G5 have potential presynaptic sites in this region, which is presumed to be a higher order, multi-sensory area and contains both arbors of descending interneurons (*Namiki et al., 2018*) and interneurons projecting to other central brain regions. Thus, these LC6G cells have the anatomy of projection neurons, not local neurons.

## Pairwise functional connectivity reveals connected downstream cell types

To assess whether the candidate downstream neuron types identified by our anatomical analysis are functionally connected to LC6s, we optogenetically depolarized LC6 neurons and measured the calcium activity of each candidate downstream cell type. In a series of pairwise experiments, the candidate downstream split-GAL4 lines drove expression of GCaMP6f (*Chen et al., 2013*), while the light-gated ion channel Chrimson (tagged with tdTomato; *Klapoetke et al., 2017*; *Strother et al., 2017*) was expressed in LC6 using the orthogonal LexA/LexAop system (*Lai and Lee, 2006*; *Pfeiffer et al., 2010*; *Figure 2A*, left, LC6 in magenta, candidate downstream LC6Gs in green; all genotypes detailed in *Supplementary file 1*). Flies were dissected and whole brains (not including the photoreceptors) were imaged using two-photon microscopy (*Figure 2A*, right). To obtain an LC6 activity-dependent tuning curve for responses of downstream neurons, we first conducted a calibration experiment to select a series of increasing light stimulation intensities that would evoke activity in LC6s with monotonically increasing responses. For this, we expressed Chrimson and GCaMP6f simultaneously in LC6 using the split-GAL4/UAS system and LexA/LexAop system. The stimulation light was spatially restricted to the glomerulus being imaged (detailed in Materials and methods), to restrict the stimulation to LC6 neurons. We established a six-pulse stimulation protocol with ramping light intensity with which we measured a monotonically increasing LC6 GCaMP signal in response to increasing levels of light-evoked Chrimson activation (*Figure 2C,D*; LC6).

Using this stimulation protocol, we measured responses from the candidate downstream neuron types while activating LC6 neurons (Region of Interest, ROI, *Figure 2B*, in cyan). All cell types, except LC26G1 (control line), showed increasing calcium responses to LC6 activation (alternative stimulation protocol and control line responses shown in *Figure 2—figure supplement 1*). The responses of the different downstream neuron types had different amplitudes and temporal kinetics (*Figure 2C*, *Figure 2—figure supplement 1A*), likely reflecting a combination of connection strengths from LC6, intrinsic properties of the target neurons, indirect contributions evoked by the stimulation, and possible expression strength differences. The amplitude difference is most clear in the response curves (*Figure 2D*, *Figure 2—figure supplement 1B*): the bilateral LC6G1 neurons showed a strikingly large $\Delta F/F$ (~9.0), whereas the other cell types showed an order of magnitude smaller increase in $\Delta F/F$ (~0.4). Part of this difference is due to the density of processes of each cell type. The LC6G1 neurons fill the glomerulus more densely than the other LC6Gs. However, this difference is unlikely to account for a ~ 20X difference in response magnitude. All cells proposed to be downstream of LC6 (*Figure 1*) showed significant responses (results of statistical analysis in *Figure 2D*, *Figure 2—figure supplement 1B*) to strong levels of LC6 activation, consistent with their being functionally connected to LC6 axons in the glomerulus.

## Neurons downstream of LC6 exhibit stereotyped spatial receptive fields

Having established multiple cell types as connected to LC6, we next examined whether these cells respond to the same visual looming stimuli that selectively activate LC6 neurons (*Wu et al., 2016*). For further analysis, we selected the strongly connected, bilaterally projecting LC6G1s as well as the

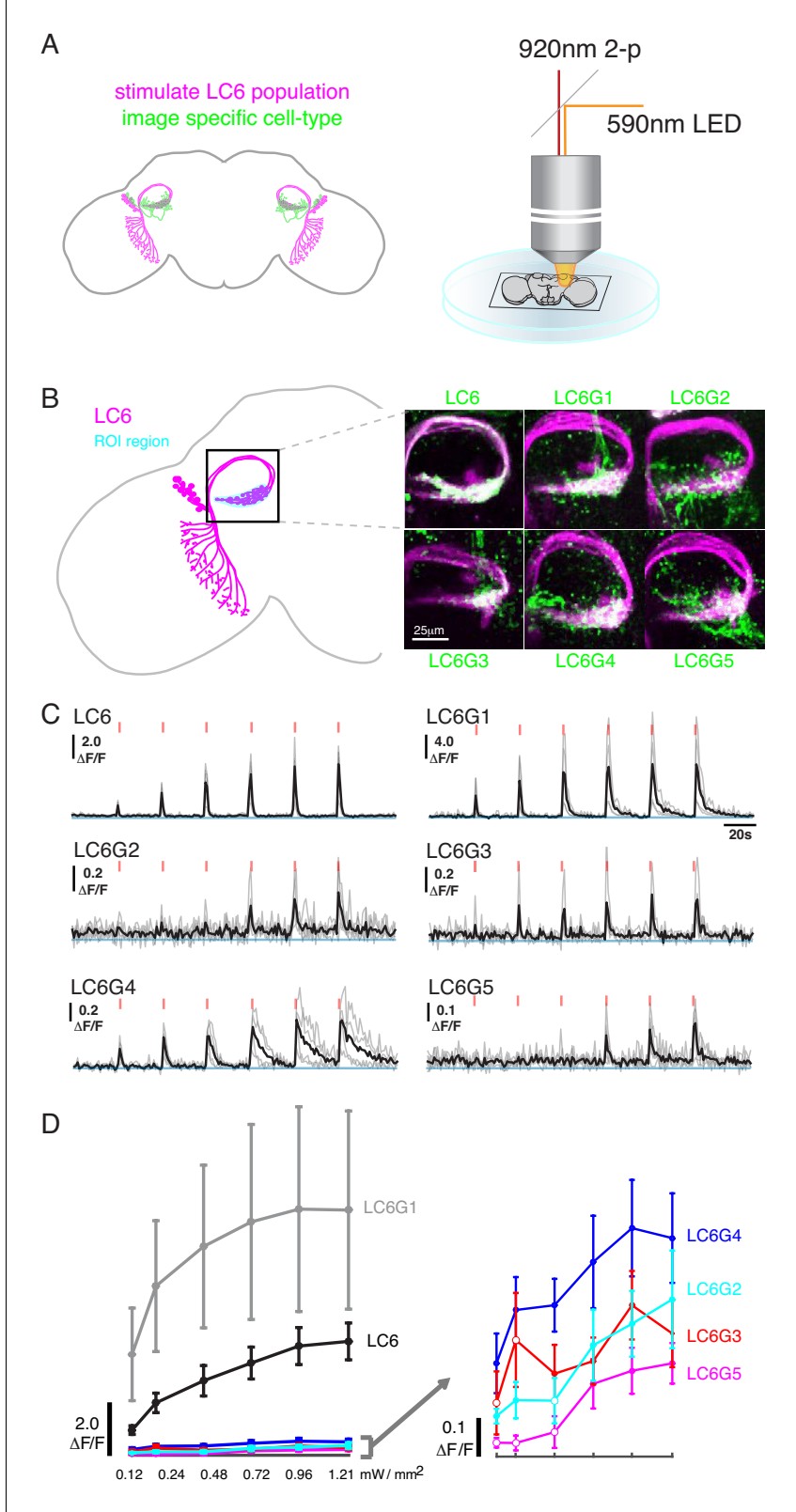

**Figure 2.** Pairwise functional connectivity reveals connected downstream cell-types. (**A**) Chrimson was expressed in LC6 (magenta) using a LexA line, and GCaMP6f was expressed in individual candidate downstream neuron types or LC6 itself (green) using split-GAL4 lines. Chrimson was activated by pulses of 590 nm light, while GCaMP responses were imaged using two-photon microscopy. (**B**) Calcium responses were measured in the ROI region in

*Figure 2 continued on next page*

*Figure 2 continued*

cyan. Representative images of double-labeled (Chrimson in magenta, GCaMP6f in green) brains used for the experiment. (**C**) Calcium responses in candidate downstream neurons in response to LC6 Chrimson activation (N = 4–5 per combination; individual sample response in gray, mean response in black). A range of response amplitudes and timecourses were observed. Red tick marks indicate the activation stimulus. (**D**) Peak responses (mean ± SEM) are shown for each candidate downstream cell type for increasing light stimulation. Closed circles denote data points significantly different from pre-stimulus baseline, open circles denote data points not significantly different from pre-stimulus baseline (p<0.1, two-sample t-test). See *Supplementary file 1* for detailed summary of fly lines used in this study.

The online version of this article includes the following figure supplement(s) for figure 2:

**Figure supplement 1.** Additional functional connectivity results with alternative stimulation protocol and an additional cell type.

LC6G2 ipsilaterally projecting neurons. We expressed GCaMP6m (*Chen et al., 2013*) using the same split-GAL4 driver lines used for the targeted functional connectivity experiments, and measured calcium responses from the LC6Gs (population activity as a summed response from ~4 labeled cells of each type, per brain hemisphere) to visual stimuli using in vivo two-photon microscopy (*Figure 3A*, left). As with LC6 neurons, both LC6G types responded to visual looming stimuli (example response for LC6G2 recordings in *Figure 3A*, middle). However, the initial measurements appeared to show substantial selectivity for stimuli presented in different spatial locations. In light of the anatomy of the glomerulus, this spatial selectivity was unexpected. To examine whether LC6G neurons consistently respond to looming stimuli in specific locations, we mapped the responses to small looming disks (maximum size 18°) presented at all locations on a grid covering a large region of the right eye's field of view (*Figure 3A*, right). We precisely measured the head orientation (*Figure 3—figure supplement 1A*) to map the visual stimulus onto the fly's eye (*Figure 3—figure supplement 1B*).

If visual-spatial information is wholly discarded in the LC6 glomerulus then the LC6G neurons would be expected to have flat receptive fields (RFs) that broadly cover the entire field of view of each eye. We found that the bilateral LC6G1 neurons responded with significant calcium increases to looming stimuli across most of the stimulated ipsilateral hemifield, while the ipsilateral LC6G2 neurons responded to looming stimuli within a more restricted portion of the visual field (*Figure 3B, C*). The strongest LC6G1 responses were to looming stimuli towards the midline and along the equator (*Figure 3B,C*, and *Figure 3—figure supplement 1C*; contours). By comparison ipsilateral LC6G2 neurons did not respond to looming stimuli near the midline, or much below the equator (*Figure 3B,C*). LC6G2 recordings from individual flies (represented as contours in *Figure 3C*, right, and as individual panels in *Figure 3—figure supplements 1C* and *2*) also show responses that are more focal, restricted to a smaller receptive field than the broader LC6G1 responses. By comparison, the responses of individual LC6 neurons, as measured in the glomerulus, are consistent with a receptive field size diameter that is not more than 20° (*Figure 3—figure supplement 3*), suggesting that LC6G1 and LC6G2 neurons integrate their inputs from many individual LC6 neurons. It is noteworthy that groups of four to sixLC6G2 neurons exhibited a similar, spatially restricted receptive field (all experiments measure the combined responses of multiple cells), indicating that the cells either share their spatial selectivity or are even more spatially selective individually. The spatial responses of both LC6G1 and LC6G2 cells are consistent across flies (*Figure 3—figure supplements 1C* and *2*), suggesting that different LC6G downstream neuron receptive fields are a robust property of each type, and therefore, spatially biased read out of the LC6 neurons is expected to be implemented in the glomerulus.

## LC6 downstream neurons differentially encode visual stimuli

While LC6G1s and LC6G2s are functionally downstream of LC6, they responded differently to visual looming stimuli based on the region of the visual field in which they are presented. Given their strikingly different morphologies, we investigated whether these neurons differentially process visual information from LC6s. We presented a panel of visual stimuli (*Supplementary file 2*) to compare calcium responses of LC6G1s and LC6G2s to those of LC6. As previously reported, LC6 preferentially responded to dark looming stimuli, and also responded to the individual features of a looming stimulus (darkening in the luminance-matched stimulus and the edge motion in the looming annulus

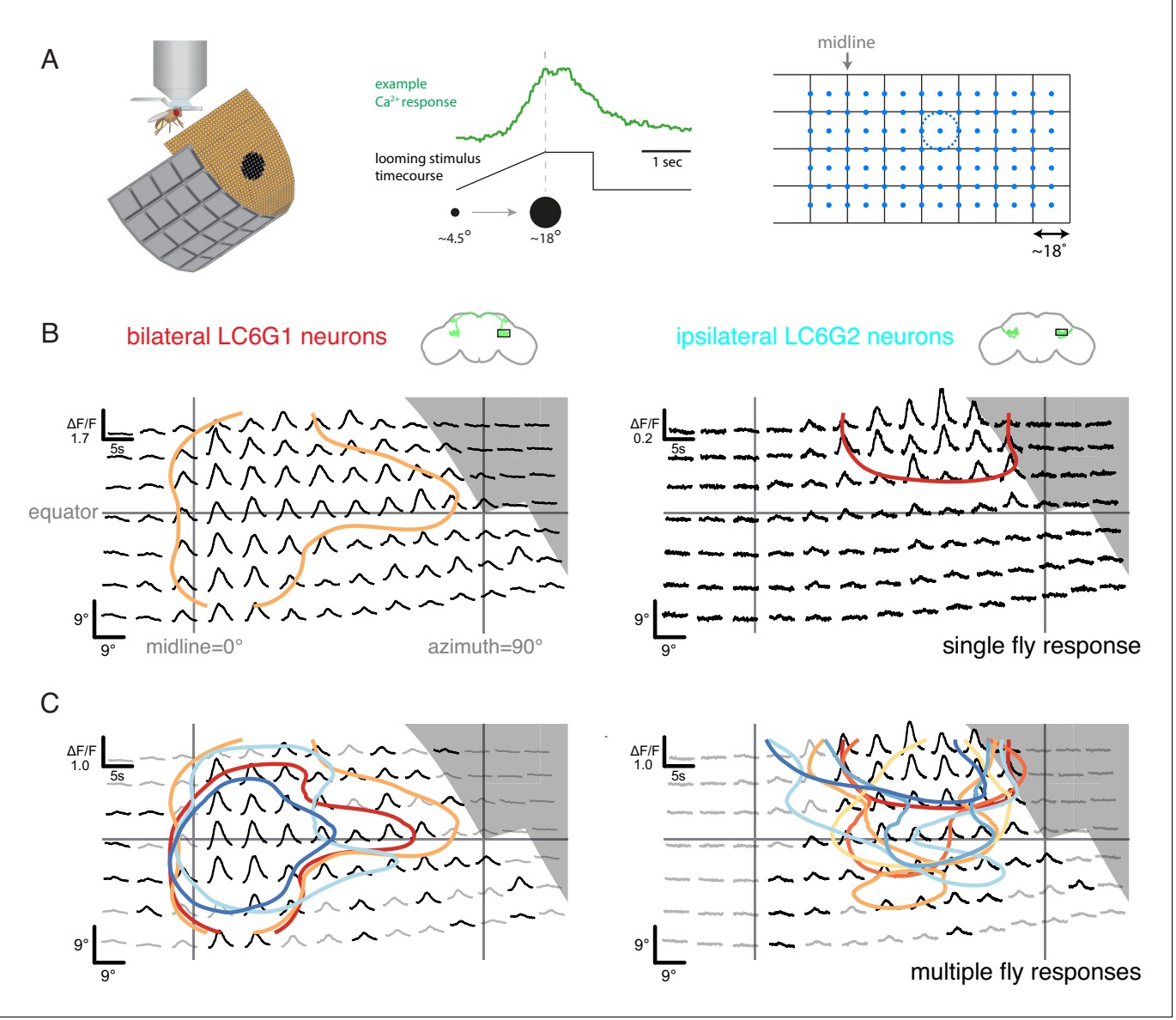

**Figure 3.** Neurons downstream of LC6 exhibit stereotyped spatial receptive fields. (A) Left: Schematic of the in vivo two-photon calcium imaging setup. An LED display was used to deliver looming stimuli at many positions around the fly, while calcium responses were measured from an ROI containing the LC6 glomerulus. Middle: Example stimulus time course and response of an LC6G2 neuron. Right: Small looming stimuli were delivered at 98 positions centered on a grid (and subtending ~ 18° diameter at maximum size) to map the receptive fields of LC6G neurons. (B, C) The bilateral LC6G1 and ipsilateral LC6G2 neurons showed consistent visual responses. LC6G1 responded throughout most of the stimulated area, with strongest responses near the equator and toward the midline, and LC6G2 responded within a more restricted receptive field. Gray-shaded regions indicate portions of the display that should not be visible to the fly. (B) Single animal example responses (from groups of ~ 4 cells of each LC6G neuron type), with each time series showing the mean of three repetitions of the stimulus at each location. The responses are plotted to represent the grid of looming stimuli mapped onto fly eye centered spherical coordinates (see *Figure 3—figure supplement 1B*; lines indicating the eye's equator, frontal midline, and lateral 90°, are shown in gray). The (0,0) origin of each time series is plotted at the spatial position of the center of each looming stimulus (this alignment onto eye coordinates accounts for e.g. the upward curving stimulus responses below the eye equator). The 60% of peak response is demarcated with a contour (based on interpolated mean responses at each location). (C) The mean responses across multiple animals are shown (N = 4 for LC6G1, N = 7 for LC6G2), with the 60% of peak contours shown in a separate color for each individual fly. The responses have been normalized for each fly before averaging (detailed in Methods). Responses at each position that are significantly larger than zero are in black, others in gray (one-sample t-test, p<0.05, controlled for False Discovery Rate *Benjamini and Hochberg, 1995*). The individual responses are shown in *Figure 3—figure supplements 1* and *2*; the gray area indicates the region of the visual display that was occluded by the head mount (*Figure 3—figure supplement 1B*). See *Supplementary file 1* for detailed summary of fly lines used in this study and *Supplementary file 2* for summary of all visual stimuli presented.

*Figure 3 continued on next page*

*Figure 3 continued*

The online version of this article includes the following figure supplement(s) for figure 3:

**Figure supplement 1.** Eye map and individual fly receptive fields.
**Figure supplement 2.** Receptive fields measured from individual flies, replotted as a heat map.
**Figure supplement 3.** Estimation of LC6 receptive field size from GCaMP imaging of putative LC6 boutons in the glomerulus.

stimulus), but not to a receding dark stimulus or a bright looming stimulus (*Wu et al., 2016*). LC6 neurons also responded to the motion of non-looming objects (*Figure 4B*; response time series shown for slowest speed, tuning curves on the right for other speeds). The bilaterally projecting LC6G1 neurons responded to the looming related stimuli as well as the bar and small object motion stimuli in a manner that was very similar to the LC6 population (*Figure 4B,C*). This similarity of LC6 and LC6G1 responses extended to all stimuli presented, which is well illustrated by the scatter plot comparing the responses of the two cell types (*Figure 4C*, left). The regression line relating (normalized) LC6 to LC6G1 responses is close to the identity line (slope = 1.05, Pearson's correlation r = 0.87), suggesting that LC6G1 may serve as a 'summary' of the LC6 populations' visual responses across all stimuli. By contrast, the ipsilaterally projecting LC6G2s were much more selective for looming stimuli, responding to dark looming and to the edge motion of the looming annulus, but not to the other stimuli (*Figure 4B*). The dissimilarity of responses can also be observed from the comparison scatter plot (*Figure 4C*, right), where the points are more dispersed away from the identity line, with looming stimuli showing an enhanced response (above the identity line) relative to most of the responses to non-looming stimuli (below the identity line). Accordingly, the linear regression shows a weaker correlation (slope = 0.51, r = 0.40). While LC6G1 appears to relay the LC6s' visual responses (presumably to the contralateral LC6 glomerulus) the LC6G2 neurons are more selective for looming than for non-looming stimuli.

## EM reconstruction of LC6 neurons connecting the optic lobe and the central brain

Our functional analysis suggests that LC6G neurons, especially LC6G2, encode the visual-spatial information of looming stimuli from their inputs in the glomerulus, but anatomical analysis of the LC6 glomerulus from light microscopy images does not clarify how this selectivity could be established (*Figure 1*). One parsimonious explanation is that this spatially selective readout of LC6 neurons could result from biased synaptic connectivity between LC6 neurons and their downstream targets. To test this hypothesis, we manually traced the LC6 neurons in a recently completed serial section transmission electron microscopy volume of the full adult brain (*Zheng et al., 2018*). We found 65 LC6 neurons on the right-hand side of the brain (single example in *Figure 5A*), which we confirmed based on their morphology in the lobula, the distinctive axonal loop tract (shared with LC9), and the orientation of the axons in the glomerulus (*Figure 5B*). We then reconstructed all LC6 neurons, completing their axon terminals (and the majority of their synapses) in the glomerulus, and tracing the major, but not the finest neurites in the lobula (*Figure 5C*). Tracing the major dendritic branches in the lobula enabled us to computationally estimate the corresponding region of the visual field for each LC6 neuron (*Figure 5D* and Materials and methods). LC6s cover almost the entire visual field with a higher density near the visual midline (*Figure 5D* and *Figure 5—figure supplement 1*). Using this mapping, we constructed 'anatomical' receptive fields of each LC6 (65 total), an estimate based only on the dendritic location and their averaged extent (*Figure 5D*, two example receptive fields are filled in, *Figure 5E*).

While no clear organization of the LC6 projections into the glomerulus was detectable in light microscopic images (*Figure 1*), the preservation of neuron identity and the higher resolution of the EM-reconstructed data set allowed us to re-examine the organization of LC6 axons. To visualize the relative organization of individual LC6 neurons, we focused on four groups, each comprised of five LC6 neurons and originating from one edge of the lobula (*Figure 5E*). Together these four groups of neurons sample the extreme points, near the boundary, of the fly eye's field of view. *Figure 5F* shows the positions of the axons of each neuron visualized at several cross-sections between the lobula and the glomerulus, as well as inside the glomerulus. While in some instances the axons from each group are nearby others of the same group (indicated by the color-code), there is substantial

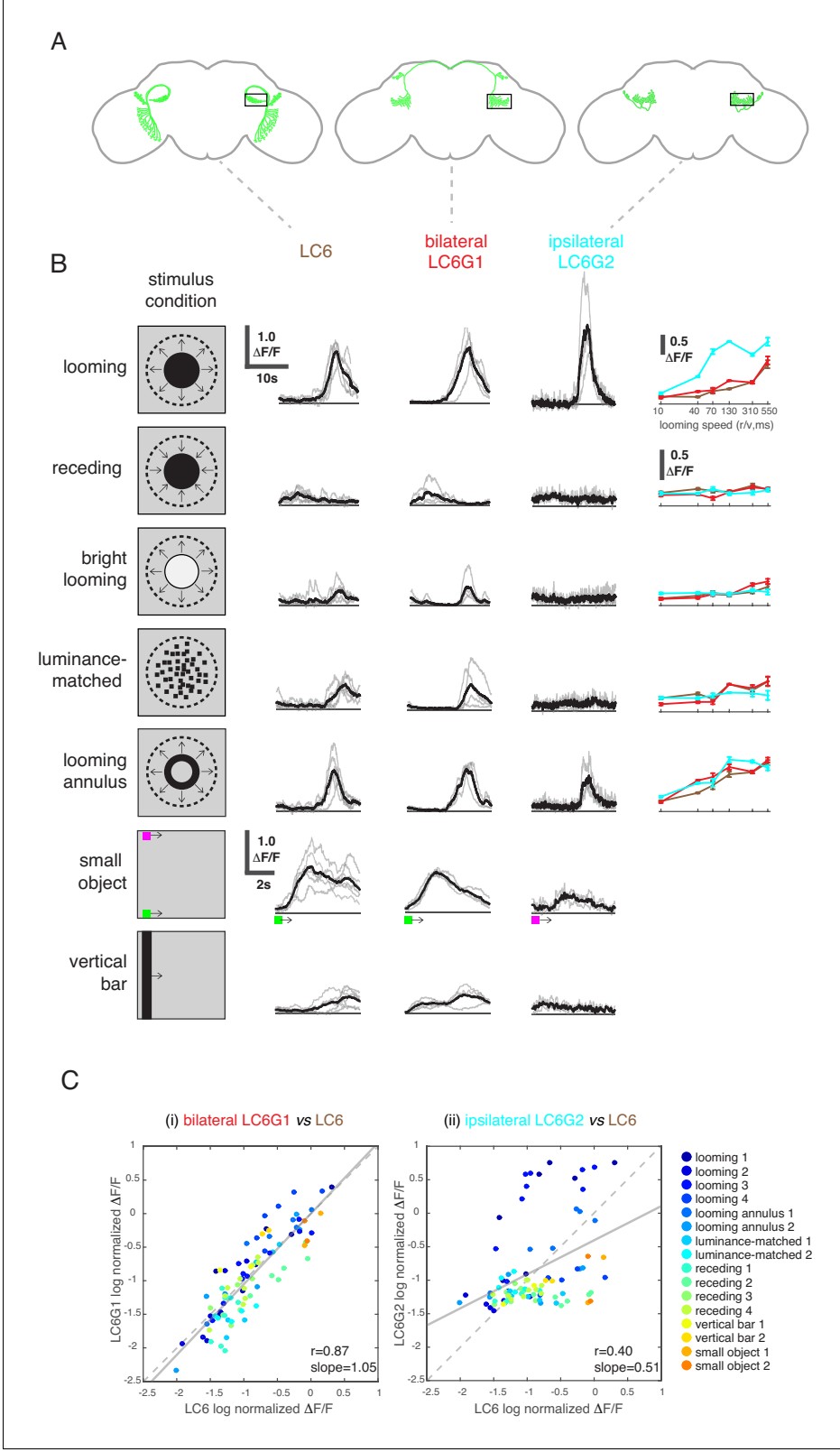

**Figure 4.** LC6 downstream neurons differentially encode visual stimuli. (**A**) Schematic of LC6, bilateral (LC6G1), and ipsilateral (LC6G2) projecting downstream cell types. Black rectangles indicate the ROI selected for GCaMP imaging; visual stimuli were presented to the ipsilateral eye. (**B**) The calcium responses in LC6 and downstream neurons evoked by visual stimuli indicated by 'stimulus condition.' Times series are shown for the lowest speed (r/

*Figure 4 continued on next page*

*Figure 4 continued*

v = 550) in each stimulus category. The small object and vertical bar stimuli moved at 10°/s. Individual fly responses are in gray, and mean responses in black (LC6 N = 6, LC6G1 N = 4, LC6G2 N = 3). The tuning curves show peak responses for each speed of the looming and looming-related stimuli (mean ± SEM). Note that 'luminance-matched' stimulus does not contain looming motion, but the overall luminance was matched, frame-by-frame, to the looming stimulus at each speed tested. (**C**) Comparison between LC6 and the LC6G visual responses. LC6 and LC6G peak responses are log normalized and plotted against each other. LC6G1 (bilaterally projecting downstream) responses correlate well with that of LC6 (Pearson's correlation, slope = 1.05, r = 0.87, p=5.7 × 10$^{-29}$), while LC6G2 (ipsilaterally projecting downstream) are not as well correlated (slope = 0.51, r = 0.40, p=7.7 × 10$^{-5}$). Note that most of the looming responses are above the (X = Y) identity line, while most of the non-looming responses are below it, demonstrating enhanced selectivity for looming stimuli. Linear fit is in solid gray line, while the identity line is in dashed gray. The data points are color coded by visual stimulus condition (detailed in Materials and methods and *Supplementary file 2*). See *Supplementary file 1* for detailed summary of fly lines used in this study and *Supplementary file 2* for summary of all visual stimuli presented.

mixing between the groups, and importantly, the set of colored axons do not define an organized set of extreme points along a boundary at any location. The relative positions of the axons of the LC6 neurons suggest that the regular, retinotopic organization present in the lobula is not directly 'inherited' by the glomerulus. The axonal projections are intermingled upon leaving the lobula (and split into separated tracts, between and within which, many other projection neurons are found, *Figure 5F* location 1) and do not exhibit clear organization in the glomerulus. And yet some preservation of the relative positions in the lobula appears in the glomerulus (*Figure 5F*, location 4), but how different is the organization at any of these locations from what might be expected due to a random re-organization of the neurons?

To explore how the organization of LC6 neurons outside of the visual system relates to their organization within the lobula, we developed a quantitative measure of 'orderliness' that when compared to a retinotopic structure, enables an assessment of how much retinotopy is present. Our measure is based on an intuitive extension of retinotopy: we look at whether neurons that are neighbors in one part of the brain are nearby in another. The simple example in *Figure 5G* illustrates the approach for a set of 4 points, that could, for example, represent the LC6 neuron positions in the lobula. We then examine the distances of a transformed set of points in another space, which could be the positions of the axons at some plane in the central brain. We first order the points in the original space on the left based on their relative distance (a is closest to b, then c, then d; b is closest to c, then a, etc.). If in the mapping from one space to another, the set of points has been rotated, but all relative distances remain the same (rotation, top), then the distance-ordered lists are identical. If a pair of 'nearby' points happen to be switched by this mapping (a/c switch, middle) then the ordering will differ from the original. Using each point as a reference, the ordered list for both spaces are permutations of each other, and we count the number of nearest neighbor swaps (inversions) required to transform one list into the other (this swapping procedure is the basis for the classic Bubble sort algorithm in computer science). In the final example, a (slightly) more distant set of points (c/d switch, bottom) is switched through the mapping, while the others remain in place. In this case the ordering is even more different, and more swaps are required to re-establish the original ordering. As the number of swaps is directly related to how much scrambling has occurred between the point sets in the two spaces, we use it as the basis of a 'Retinotopy Index' (RI) that is defined for the i$^{th}$ point as $RI_i = 1 - \frac{S}{A}$, where S denotes the number of swaps required to transform one list into the other, and A denotes the expected value of the number of swaps for a set of points ($A = \frac{(N-1)(N-2)}{4}$, for N points, including the reference point; see Materials and methods for further details). The index for the mapping of the population is defined as $\bar{RI}$, which is the average RI over all points. In the plotted data in *Figures 5* and *6*, the population $\bar{RI}$ is clear from context, and so we refer to this quantity simply as RI. RI is 1 for mappings that do not re-order the set of points, -1 for a mapping that inverts all distance relationships, and is on average 0 (since S = A) for random mappings (right side of example in *Figure 5G*). In *Figure 5H*, we apply this analysis to an example more directly related to the LC6 mappings. For a set of 66 points, if the mapping maintains much of the global organization but scrambles local positions (similar to how retinotopically organized neurons in the fly optic lobe project through the second optic chiasm *Shinomiya et al., 2019a*), the RI is reduced from

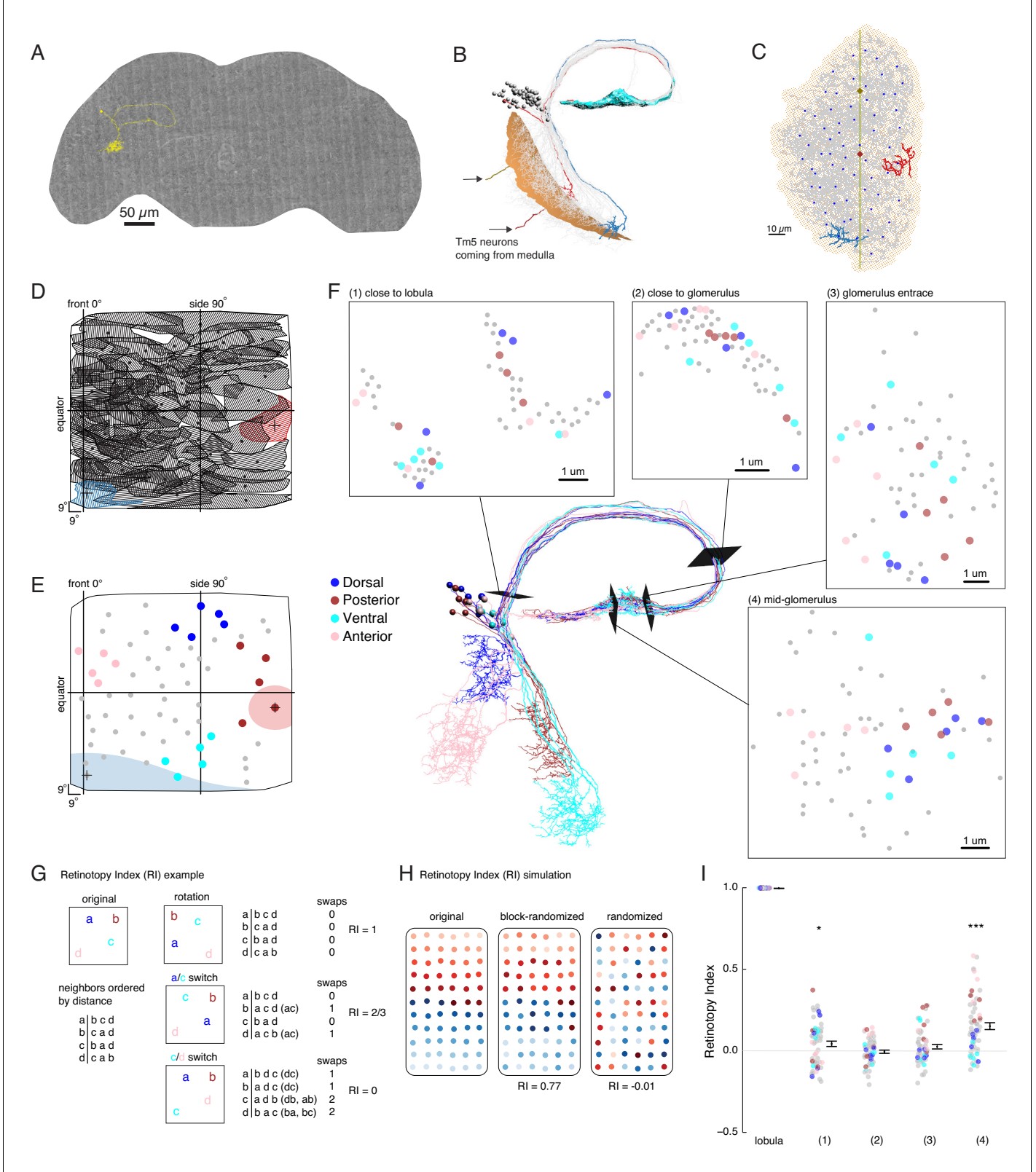

**Figure 5.** EM reconstruction of LC6 neurons connecting the optic lobe and the central brain. (**A**) An example LC6 neuron found in Serial Section Transmission Electron Microscopy volume of the complete female brain. (**B**) All LC6 neurons on one side of the brain (by convention, the right side, but displayed here on left) were traced into the glomerulus. Two example cells are highlighted in red and blue (same cells in **C**, **D**, and **E**). All cells have dendrites mainly in lobula layer 4 (orange), and project to the LC6 glomerulus (cyan). Cell bodies are rendered as gray spheres. The projections of two

*Figure 5 continued*

Tm5 neurons are shown coming from the medulla side of the lobula. They were selected to identify the center (brown Tm5) and central meridian (both Tm5s) of the eye in C. (C) Projection view of the lobula layer. Traced LC6 dendrites (gray) are projected onto a surface fit through all dendrites (light orange). Blue disks represent centers-of-mass of each LC6 dendrites. The vertical line is the estimated central meridian, the line that partitions the eye between anterior and posterior halves, mapped onto the eye coordinate in D. (D) Estimate of dendritic field coverage of visual space (basis for anatomical receptive fields) for all LC6 neurons. Polygons are the estimated visual fields, and the boundary indicates the estimated boundary of the lobula (layer 4). The red example cell RF corresponds to posterior-viewing parts of the eye while the blue example cell RF corresponds to fronto-ventral viewing direction. In agreement with previous characterization of LC6 neurons (*Wu et al., 2016*), the dendrites appear to cover the entire visual field with significant overlap. This overlap is most prominent in the region of the lobula corresponding to the fronto-ventral part of the eye. Note that the central meridian is not shown in D and is not exactly the same as the line indicated by 'side 90°'. (E) Four groups of five LC6 neurons, corresponding to different regions of visual space, used for exploring retinotopy in F. The filled regions are examples of the anatomical receptive field estimates, for the red and blue neurons in (B), (C), and (D). (F) Central figure shows the skeletons of the four groups of LC6 neurons with four cross-sections: (1) outside the lobula, (2) before the entrance of the glomerulus and (3 , 4) within the glomerulus. The disks in the four cross-section plots represent LC6 axons with the same color code as in (E). The positions of these axons show significant mixing, and do not generally mark the boundary of the group of neurons, as they do in the Lobula. The gray dots correspond to the 45 LC6 neurons not identified by one of the four groups; for compactness of this visualization, up to one gray dot per panel is omitted. (G) An example illustrating the Retinotopy Index, RI, that quantifies the amount of rearrangement occurring when a set of points is mapped from one space to another. On the left side is an example for four points, in the original space (such as the lobula), below is the (Euclidean) distance-based ranking of the points, using each point as a reference. In the middle are example transformations and the resulting distance-ranked lists. As detailed in the text and in the Materials and methods, the quantification is based on counting the number of swaps required to re-order the transformed list of points back to the original list. The RI for a point set and mapping is the averaged RI over all reference points. For mappings that preserve the order, the RI will be close to 1, while RI ~ 0, corresponds to (on average) a random mapping of the points (and therefore a random re-ordering of retinotopy). (H) Simulated examples of 2 mappings applied to a (jittered) grid of 66 points. In the first example ('block-randomized'), the mapping maintains most of the global organization, but within the local blocks, the order has been randomized, resulting in an RI that is close to, but reduced from 1. In the second example, the order of the points has been completely randomized and the resulting RI for this mapping is ~ 0. (I) Retinotopy Indices for all 65 LC6 neurons (shown as individual points) at four cross-sections defined in (F) (plotted with their corresponding colors), with the lobula being the original space (RI = 1 at that location). The black dot and horizontal bars denote the mean ± SEM. Asterisks represent the result of the Mann-Whitney U test, comparing the population to 0; *p<0.05, ***p<0.001.

The online version of this article includes the following figure supplement(s) for figure 5:

**Figure supplement 1.** Distribution of LC6 arbors in the lobula from EM data.

**Figure supplement 2.** Retinotopy Index for simulated examples.

1 (0.77), but is still very different from a completely randomized mapping, for which the RI ~ 0. A larger set of simulated mappings is considered in *Figure 5—figure supplement 2*. This quantification has many convenient features: it accounts for the organization of all neurons in the set, does not impose any expectation for how the neurons should be ordered (no requirement for regular tiling, etc.), and can be applied to a range of mappings, so long as a distance metric is defined (not just 2D to 2D).

We then applied this quantification to the LC6 axon positions at the locations (two along the axon tract and two inside the glomerulus) in *Figure 5F*. The RI shows very little organization as compared to the lobula at any position. While the RI is significantly different from zero just outside the lobula (position 1), at position 2 (axon tract) and 3 (glomerulus entrance), the population RI is indistinguishable from a completely randomized mapping. Intriguingly, in the middle of the glomerulus (position 4, which was chosen because of the relatively high RI compared to other glomerulus locations) the axons of LC6 neurons are significantly more organized (with respect to their lobula layout) than is expected from a random mapping. However, this quantification, as well as the qualitative inspection of the set of points in *Figure 5F*, position 4, suggests that the packing of neurons within cross-sections of the glomerulus exhibit only a minimal reflection of their organization in the lobula. This high-resolution examination of the LC6 axons reveals a limited signature of retinotopic organization in the glomerulus. Two outstanding question remain: is there further organization in the glomerulus? And does anatomy reveal whether LC neurons or their downstream targets neurons make use of this structure?

## The organization of LC6 axons and their synapses in the glomerulus

To address the first question, we looked for spatial biases within the glomerulus by comparing the median positions of each LC6 axon terminal. We calculated these median positions based on either the reconstructed skeleton, or from the positions of the annotated pre-synaptic sites (*Figure 6A*),

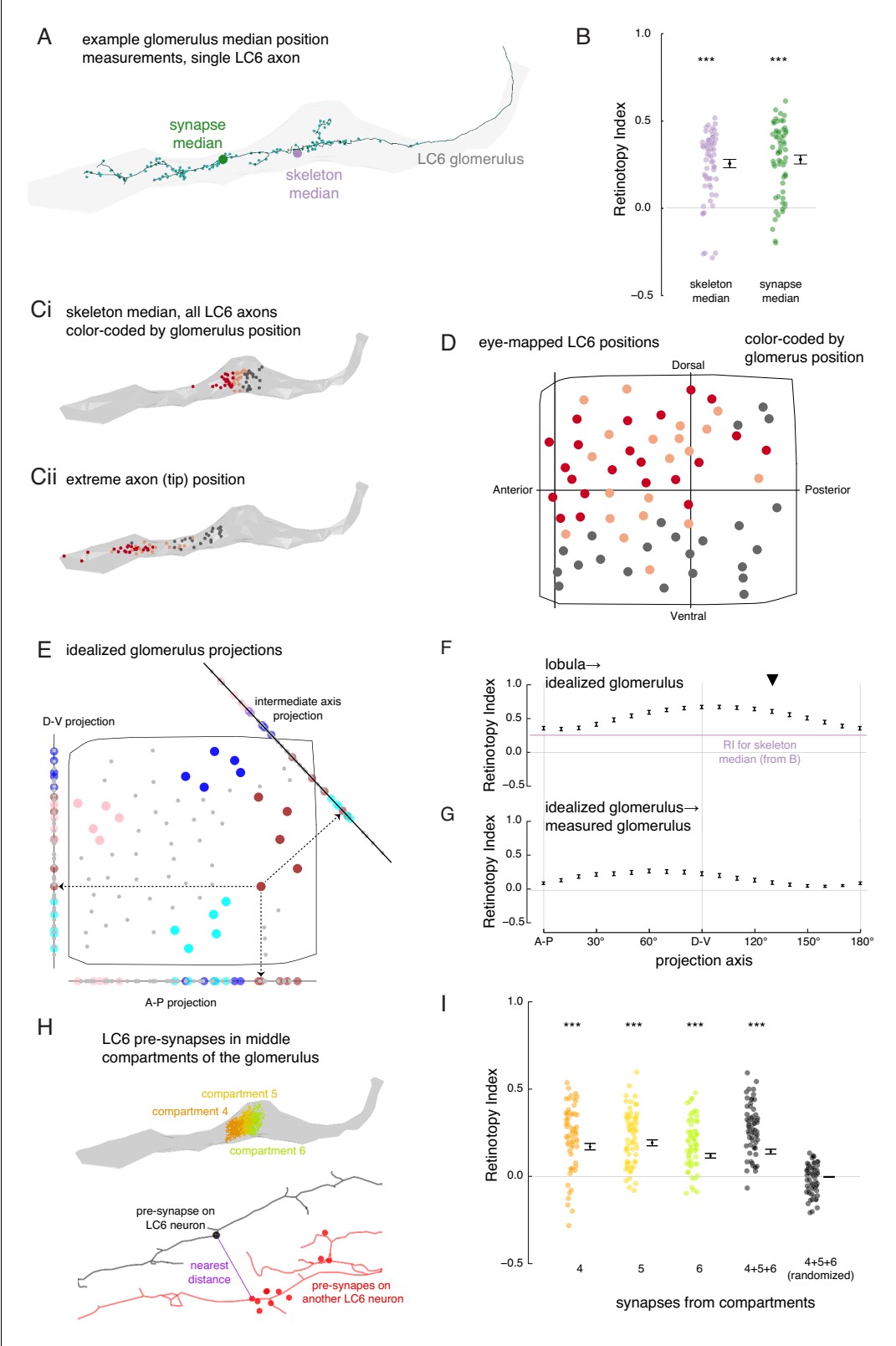

**Figure 6.** The organization of LC6 axons and their synapses in the glomerulus. (**A**) An example of an LC6 axon within the glomerulus (in gray), with pre-synaptic sites indicated as cyan dots. The purple disc represents the median position of the skeleton while the green disc represents the median position of the pre-synapses (all positions first projected onto the long axis of the glomerulus). (**B**) Retinotopy Index for the mappings from lobula (as in *Figure 5D*) to the skeleton median and synapse median positions (summary is mean ± SEM). Asterisks represent the result of the Mann-Whitney U test,

*Figure 6 continued on next page*

*Figure 6 continued*

comparing the population to 0; ***p<0.001; plotting conventions as in *Figure 5I*. (**Ci**) LC6 skeleton median positions (illustrated in **A**) partitioned into three groups based on position on the long axis of the glomerulus (in gray). (**Cii**) The axon tip positions (the average position of the 10% of the nodes farthest from the glomerulus), color-coded based on the groups in **Ci**, reveal that the small differences in axon median positions reflect differences in the length of the axon. (**D**) LC6 dendrite center-of-mass in the visual field (based on *Figure 5D*) color-coded by their median position in the glomerulus (**Ci**). (**E**) As a comparison for the observed lobula-to-glomerulus position mapping, we construct a synthetic mapping as the projection from the 2D lobula space onto a 1D line-space, the 'idealized glomerulus', which is swept through a range of angles. Three example 1D line-spaces are shown along with the projection of the set of points representing the LCs onto each line; neurons are color-coded as in *Figure 5E* to help illustrate the transformation. (**F**) The Retinotopy Index (mean ± SEM) for the synthetic mapping from the lobula to the idealized glomerulus. The three line-spaces shown in **E** are labeled with 'A-P', 'D-V' and the intermediate axis is indicated with a black triangle. The mean RI corresponding to the skeleton median (from **B**) is reproduced. (**G**) The Retinotopy Index for mappings from the idealized glomerulus line-spaces to the skeleton median in the glomerulus. (**H**) Three compartments in the middle of the glomerulus, selected since they contain synapses of all LC6 neurons; the colored dots show the position of the pre-synapses on all LC6 neurons in each compartment (*Figure 6—figure supplement 1B*). Below is an example illustrating the key step in a synapse-based definition of distance: the minimal distance between a pre-synapse on one LC6 neuron and the set of pre-synapses on a second LC6 neuron. See Materials and methods for details. (**I**) The Retinotopy Index (summary is mean ± SEM) for comparing the synapse-based distance of the LC6 neurons to their ordering in the lobula, for each of the three compartments in **H**, all three compartments combined, and combined then identity-randomized. Asterisks represent the result of the Mann-Whitney U test, comparing the population to 0; ***p<0.001; plotting conventions as in *Figure 5I*.

The online version of this article includes the following figure supplement(s) for figure 6:

**Figure supplement 1.** Biased connectivity and visual sampling by LC6 synapses in the glomerulus.

projected onto the 'long-axis' of the glomerulus. For each of these point sets, we then scored the Retinotopy Index (with respect to the lobula positions). The RI for both the skeleton and synapse-based median positions are similar, and in both cases significantly different from a random distribution (RI >0, *Figure 6B*). This suggests that there is some retinotopic organization along the long axis of the glomerulus. To see if there is any simple organization, we sorted the LC6 neurons based on their median position in the glomerulus to visualize their corresponding lobula positions (*Figure 6Ci, D*). From this partitioning of the LC6 neurons into three groups, we see that there is a rough correspondence between an LC6 neuron's dorso-ventral position in the lobula and the position of its axon terminal in the glomerulus. The median positions of all LC6 neurons are close to the middle of the glomerulus (*Figure 6Ci*), but they reflect the fact that some LC neurons have axons that extent towards the tip of the glomerulus (and are likely to correspond to the more dorsal portion of the eye), while others have shorter axons (corresponding to more ventral eye regions) that terminate closer to the middle of the glomerulus (visualized by axon tip position, *Figure 6Cii*).

Since LC6 neurons corresponding to different parts of the eye preferentially terminate in different parts of the glomerulus, we expected to find further evidence for the preservation of visual-spatial organization in the glomerulus. We examined this proposal by analyzing the connections between LC6 neurons that we discovered while reconstructing their axons. LC6 neurons are prominently connected to each other in the glomerulus (1902 total synapses, a mean of ~29 synapses to each LC6 neuron from all other LC6 neurons, *Supplementary file 3*). We used this connectivity to look for a correlate of the retinotopic organization in the glomerulus: do LC6 neurons preferentially synapse onto LC6 neurons with nearby lobula positions? When the distance between RF centers (measured across the lobula) was weighted by the number of synapses between the LC6 cells, we found that the average distance between connected neurons is significantly smaller than the distances after random shuffling of these connections (*Figure 6—figure supplement 1A*). Therefore, LC6 neurons bias their connectivity, such that they connect to their lobula neighbors with a distribution significantly smaller than expected for random connectivity.

Since LC6 axons preferentially synapse onto other LC6 cells with neighboring dendrites, we next examined whether these synapses exhibit any spatial organization. To visualize this, we divided the glomerulus into 10 compartments along its long axis, each containing the same number of LC6 pre-synapses (across all postsynaptic cells; see 'computational analysis of EM reconstruction' in Materials and methods; *Figure 6—figure supplement 1B*). We then examined the spatial distribution of the LC6 input within each compartment as a contour plot in visual coordinates (*Figure 6—figure supplement 1C*). We found that the most lateral glomerulus compartments (1 - 4) are biased toward representing the dorsal visual field (with a strong bias toward antero-dorsal in compartment

1). These biases are consistent with the eye-mapped LC6 positions in *Figure 6D*. Most of the remaining compartments sample visual inputs more broadly, with peak responses close to the center of the eye (consistent with broad inputs covering much of the visual field; *Figure 6—figure supplement 1C*). This suggests that the more medial compartments (e.g. 8–10) feature synapses from a broad array of LC6 neurons and are not heavily biased toward the 'shorter axon' neurons of *Figure 6C,D*, which would manifest as more ventrally biased receptive fields. Whether further retinotopic readout is supported by connectivity within the compartments is explored in *Figure 6H,I*.

While we find a clear visual-spatial bias based on the median positions of LC6 axons in the glomerulus (*Figure 6D*), how well does a simple projection from the (2-D) lobula eye-map onto the (1-D) long axis of the glomerulus capture the transformation? To understand how much order is captured by the measured RI score, we consider 'idealized glomerulus' mappings in which the position of each LC6 neuron is determined by its projection onto a primary axis in the lobula (*Figure 6E*). Then we compared the RI based on the mapping from the lobula to the idealized glomerulus, swept through projection axis angles, and also compare to the measured RI for the real glomerulus (*Figure 6F*). These comparisons show that the observed mapping (for the skeleton median) is close to the lower end of the range obtained by the synthetic mappings, indicating that even this simple projection leads to a stronger preservation of retinotopy than we measured for the actual mapping. For example, a projection along the D-V axis yields an RI that is significantly larger (p<1e-11). The dependence of RI on the projection axis angle is in part due to the anisotropic distribution of points. There are more points, and thus more spatial variance, along D-V than there are along A-P, and thus the RI is higher for projections onto the D-V axis (example for the simulated point set in *Figure 5—figure supplement 2*). But how well does this idealized mapping match the real mapping? We again applied the RI analysis, by comparing the order of LC6 neurons in each synthetic mapping to their order along the glomerulus (*Figure 6G*). We find that the preserved retinotopy is highest for axes around the D-V axis (with the maximum RI for the 60° projection axis, and a minimum for the orthogonal axis around 150°), but we also find that this projection always results in an RI <0.5, suggesting that the simple 1-D projection, which correlates with the median position along the glomerulus, is nevertheless not a very accurate approximation of the mapping from the lobula into the glomerulus.

The preceding analysis shows some visual-spatial organization at the scale of the entire glomerulus (*Figure 6D*) that is only somewhat captured by a projection onto any single axis. It is therefore apparent that downstream neurons could not extract strong spatial information, and thus a focal RF, by indiscriminately sampling LC6 inputs from a few compartments of the glomerulus (with the possible exception of the lateral 'tip' compartments, *Figure 6—figure supplement 1C*). To ask whether a finer-scale organization is present, that would be directly relevant to integration by downstream neurons, we next examined the spatial organization of the distribution of LC6 pre-synapses, and specifically examined them in the middle compartments of the glomerulus (*Figure 6H*; compartments defined to each contain ~10% of pre-synapses across the 65 LC6 axons, *Figure 6—figure supplement 1B*). The middle compartments appear to represent the most substantial challenge for the formation of spatially restricted receptive fields by downstream neurons since all LC6 axons (long and short) are present in these compartments, and the LC6 pre-synapses in each compartment correspond, as an ensemble, to broad regions of the eye (*Figure 6—figure supplement 1C*). To establish any finer-scale organization, we asked whether individual pre-synapses of LC6 neurons in the middle compartments are close to pre-synapses from LC6 neurons that are close in the lobula. We defined a pre-synapse-based distance metric for any pair of LC6 neurons. For a given pre-synapse on the first LC6, we searched for the closest pre-synapse on the second LC6 (example in *Figure 6H*, bottom). We repeated this for *all* pre-synapses of *both* neurons and define the mean as the pre-synapse distance between these two LC6s. Using this distance metric, we calculated the RI for the pre-synapses within compartments 4, 5, and 6, as well as all pre-synapses in the three compartments at once, and compared this to a randomized control where the neuron identity were randomized (*Figure 6I*). This analysis shows that the LC6 pre-synapses in the middle of the glomerulus exhibit significantly nonrandom spatial organization with RI >0 (and with no significant difference between the RI for any of the compartments, or for all pre-synapses in these compartments treated as one group), while the randomly permuted data set shows, expectedly, an RI that is indistinguishable from zero. However, the RI we find here, while non-zero, is low (~0.2), suggesting a clear signal of retinotopy whereby the output sites of the LC6 axons reflect some of the spatial organization of the visual system, but it is

far coarser than what is typically seen in e.g. columnar circuits of the optic lobes (for an example see Figure 2A–C of *Nern et al., 2015*).

## Connectivity-based readout of visual information in the LC6 glomerulus corroborates features of the spatially selective responses of target neurons

The presence of some retinotopic organization in the glomerulus, suggests that the readout of LC6 inputs may account for the spatially selective responses of the LC6G neurons. We manually reconstructed neurons within the LC6 glomerulus (by following, at random, neurites postsynaptic to LC6s), until we could match reconstructed neurons to the bilateral LC6G1 and ipsilateral LC6G2 cell types that were examined in *Figures 3* and *4*. We performed our analysis on the processes of four bilateral LC6G1 neurons (two each, per side of the brain) and on the arbors of five unilateral LC6G2 neurons. These nine neuronal arbors were fully reconstructed and independently proofread (in the glomerulus; examples in *Figure 7A*). The connectivity between these 9 arbors and the 65 LC6 neurons is summarized in *Figure 7B* (and detailed in *Supplementary file 3*). As expected from the morphology of these LC6 target neurons, and their functional connectivity (*Figure 2*), we found many synapses between the LC6 neurons and the right-hand side LC6G1 (R) and LC6G2 neurons (mean number of synapses: 912 and 428, respectively). We also found connections from LC6 pre-synapses onto post-synaptic sites on the axons of the left-hand side LC6G1 (L) neurons (mean of 207; LC6G1 (L) in *Figure 7B*); these (likely inhibitory) neurons conveying contralateral visual inputs therefore also receive ipsilateral visual inputs. Further we find a similar number (~220) of feedback synapses from the bilateral LC6G1 neurons of both sides onto LC6 cells.

With the synapses between LC6 neurons and their targets mapped, we could finally attempt to correlate the connectivity-based readout of visual information with our functional measurements of spatial receptive fields (*Figure 3*). Using gaussian approximations for the LC6 anatomical RFs (*Figure 5E*), we estimated the anatomical RFs of the individual target neurons by summing over all input LC6 RFs, scaled by the number of synapses connecting each LC6 to that target neuron. We summed the estimated RFs for each cell type across individual cells to compare with the population measurements from calcium imaging experiments (*Figure 7—figure supplement 1*, see methods). These summed estimated RFs are shown for LC6G1 and LC6G2 (*Figure 7C*) across the field of view of the entire right eye. While both estimated RFs are quite broad, there are differences between them that agree with several distinguishing features of our functional RF mapping of these cell types (*Figure 3*). To generate a more direct comparison, we aligned the functional RFs measured through in vivo calcium imaging (generated with stimuli that only partially covered the visual field; *Figure 3C*) to the 70% contour of the anatomical estimate for each reconstructed cell type (*Figure 7D*). While there is only moderate concordance between these RF measurements, there are several key points of agreement: the LC6G1 receptive field in both measures is broader and shows strong support close to the frontal midline (0° azimuth), while the LC6G2 receptive field in both measures shows support in a smaller zone away from the frontal midline. The higher density of LC6 neurons near the middle of the eye leads to rather similar peaks, along the vertical 90° line, in the EM-based predictions for both cell types. The differences between the EM-based estimates for the LC6G1 and LC6G2 receptive fields are shows in *Figure 7—figure supplement 2A,B*, where the LC6G1 functional response peaks are seen to align with the largest EM-based predictions for LC6G1 responses, while the functional LC6G2 RFs do not. Considering that these estimates of the RF of each cell type were generated with very different methods, and accounting for the approximations required to align the two data sets, we find the level of overlap between the RF measures to be substantial.

Can the visual-spatial biases in the RFs estimated for the LC6G neurons (*Figure 7C*) be explained by the organization of LC6 axons (*Figure 6*) in the glomerulus? If, perhaps during development, LC6G neurons could optimize their innervation of the LC6 glomerulus to exploit the modest retinotopic organization of the LC6 axons, then might find that LC6G neurons, on average, arborize closer to the LC6 neurons from which they get the most inputs. To look for evidence of this, we focused on two individual LC6G neurons that capture the major differences between the population averages of the EM-reconstructed cell types—LC6G1c receives more inputs from (visual) anterior LC6 neurons, while LC6G2c receives most inputs from (visual) lateral LC6 neurons (*Figure 7—figure supplement 1*). We then ask whether these LC6G neurons are closer, in the glomerulus, to the axons of LC6 cells from the anterior or the lateral visual field (see *Figure 7—figure supplement 2C,D*). We carried out

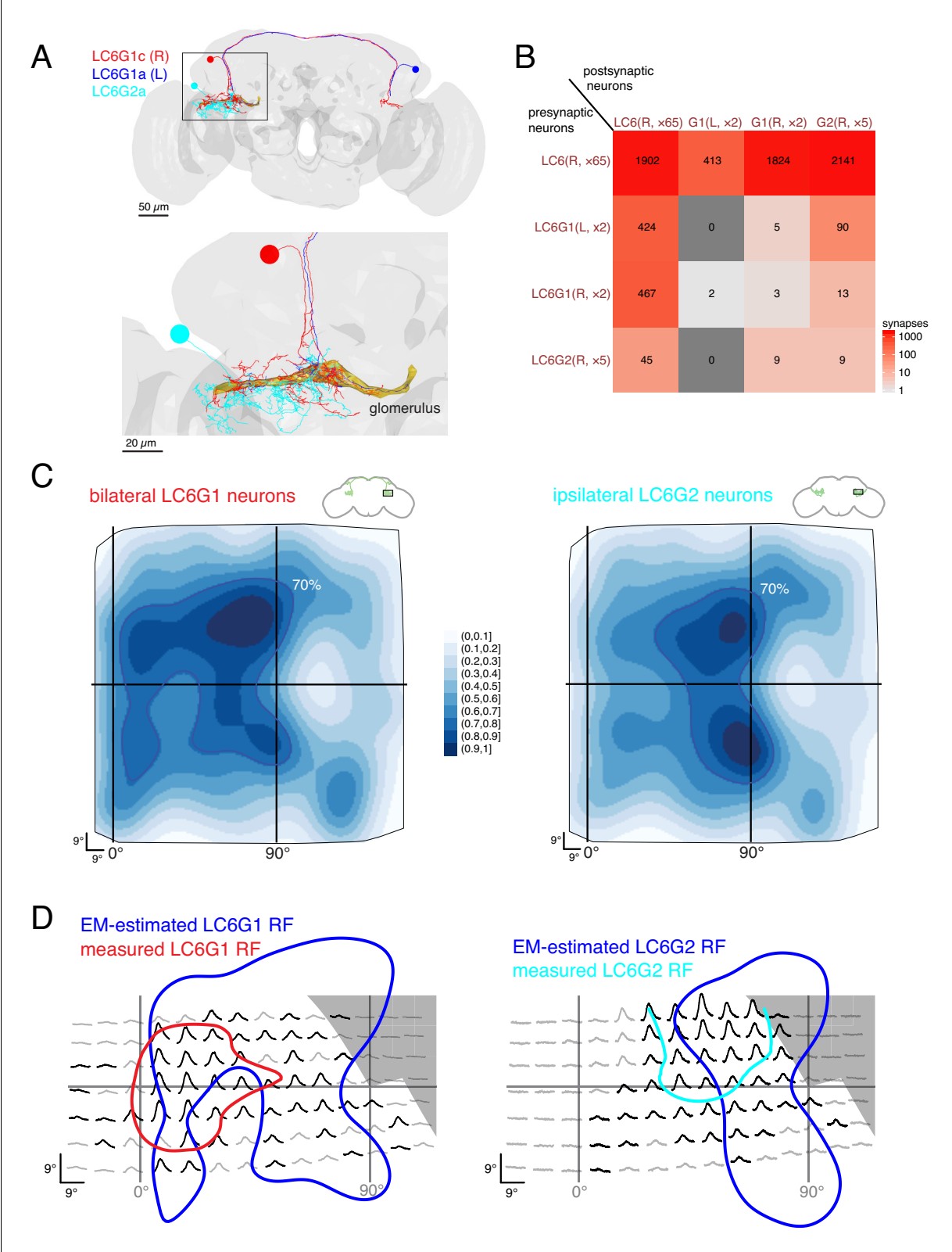

**Figure 7.** Connectivity-based readout of visual information in the LC6 glomerulus corroborates features of the spatially selective responses of target neurons. (**A**) Bilaterally projecting LC6G1 and ipsilaterally projecting LC6G2 neurons were identified and reconstructed in the full brain EM volume. Individual neurons shown in *Figure 7—figure supplement 1*. (**B**) Summary connectivity matrix of LC6, LC6G1, and LC6G2 neurons, each cell type is grouped, with the side of the brain (R or L) and the number of neurons in each group labeled. See *Supplementary file 3* for the complete

*Figure 7 continued on next page*

*Figure 7 continued*

connectivity matrix. (C) Estimated anatomical RFs of each target neuron type. Each LC6 neuron's RF is scaled by the number of synapses to each individual target neuron, and then summed across all neurons of the same type (see Materials and methods for details). RFs for each individual target neuron shown in *Figure 7—figure supplement 1*. The 70% contour is highlighted with a dark blue line. (D) Anatomical RF (blue contour) overlaid onto functional RF measured in vivo (replotted from *Figure 3C*, but with the average 60% of peak contour shown here). Further comparisons between these cell types and between EM and functional estimates of the RFs are in *Figure 7—figure supplements 2* and *3*.

The online version of this article includes the following figure supplement(s) for figure 7:

**Figure supplement 1.** Details of reconstructed LC6G neuron morphology and estimated anatomical Receptive Fields.
**Figure supplement 2.** Organization of downstream target neurons in the glomerulus.
**Figure supplement 3.** Synapse count, cable length and synapse density of downstream neurons along the glomerulus.

this analysis in thin, 1 µm sections of the glomerulus, and used individual LC6G neurons, so that any effects of proximity would not be lost due to spatial averaging, and yet we did not find any consistent difference in the distance between LC6G1c and LC6G2c and the anterior and lateral LC6 neurons (along with the remaining right side LC6G neurons also examined in *Figure 7—figure supplement 2E*). Furthermore, we find that the LC6G neurons sample the glomerulus in an approximately homogenous manner (density of synapses per unit of cable only varies by about ±50% from the mean for all neurons, *Figure 7—figure supplement 3*). We interpret these two results to mean that at the intermediate scale of LC6G arborization patterns within the glomerulus there is no trivial organization that can explain the spatially localized receptive fields. This result is further supported by the fact that the strongest bias within the glomerulus is found along the (visual) D-V axis and yet the differences between the RFs of the LC6G1 and LC6G2 neurons are primarily along the A-P axis, for which the glomerulus appears to have much less organization (*Figure 6C–G*). Taken together, this connectomic analysis of the LC6 glomerulus suggests that spatially selective readout by optic glomerulus neurons is accomplished using fine-scale, biased connectivity. The glomerulus is very crowded, with most of visual space available for sampling in nearly all locations, and therefore no simple innervation of this dense structure appears to be a sufficient mechanism to explain the formation of visual-spatial receptive fields.

## Interconnections between the LC6 glomeruli support contralateral suppression of visual responses to looming stimuli

One prominent feature of the fly nervous system is that brain regions are often connected—directly or indirectly—to their symmetric regions across the hemispheres (*Shih et al., 2015*). What role does this flow of information play? Of the LC6G neurons we identified (*Figure 1*), only the LC6G1 neurons consist of a symmetric population of neurons that cross the midline and innervate both LC6 glomeruli. The set of neurons whose interconnections we completely reconstructed within the LC6 glomerulus are fully detailed in *Supplementary file 3* and summarized in *Figure 8A*. The LC6G1 neurons from both hemispheres are both pre- and post-synaptic in the LC6 glomerulus. The right LC6G1 (R) neurons receive ~4.4 times more synapses from LC6 neurons in the right glomerulus than the left LC6G1 (L) neurons; however, these neurons from both sides of the brain provide a similar number of synapses back onto LC6 neurons. Only the contralateral LC6G1 (L) neurons synapses onto the ipsilateral LC6G2 neurons, whereas the LC6G2 neurons do not synapse onto either of the LC6G1 neurons.

By combining the connectivity of these three cell types together with the neurotransmitter expression profiles (*Figure 1—figure supplement 3*), a simple microcircuit emerges (*Figure 8A*). The flow of information from the contralateral LC6 glomerulus via the putatively inhibitory LC6G1 neurons may serve a gain control function within the glomerulus using both ipsilateral and contralateral responses of LC6 neurons to dampen strong aggregate responses from the inputs to the glomerulus. Furthermore, the contralateral LC6G1 (L) neurons directly synapse onto the ipsilateral LC6G2 (R) neurons, potentially suppressing their stimulus responses as a consequence of contralateral visual inputs.

To test this proposal, we recorded the calcium responses of the ipsilateral LC6G2 neurons while presenting looming stimuli to each eye (*Figure 8B*). As expected from the results in *Figure 4*, when a single looming stimulus was shown, LC6G2 neurons responded to the ipsilateral, but not the contralateral, presentation (*Figure 8B*, columns 1,2). When both sides were simultaneously stimulated with looming, the responses were significantly reduced at all tested speeds (as compared to the

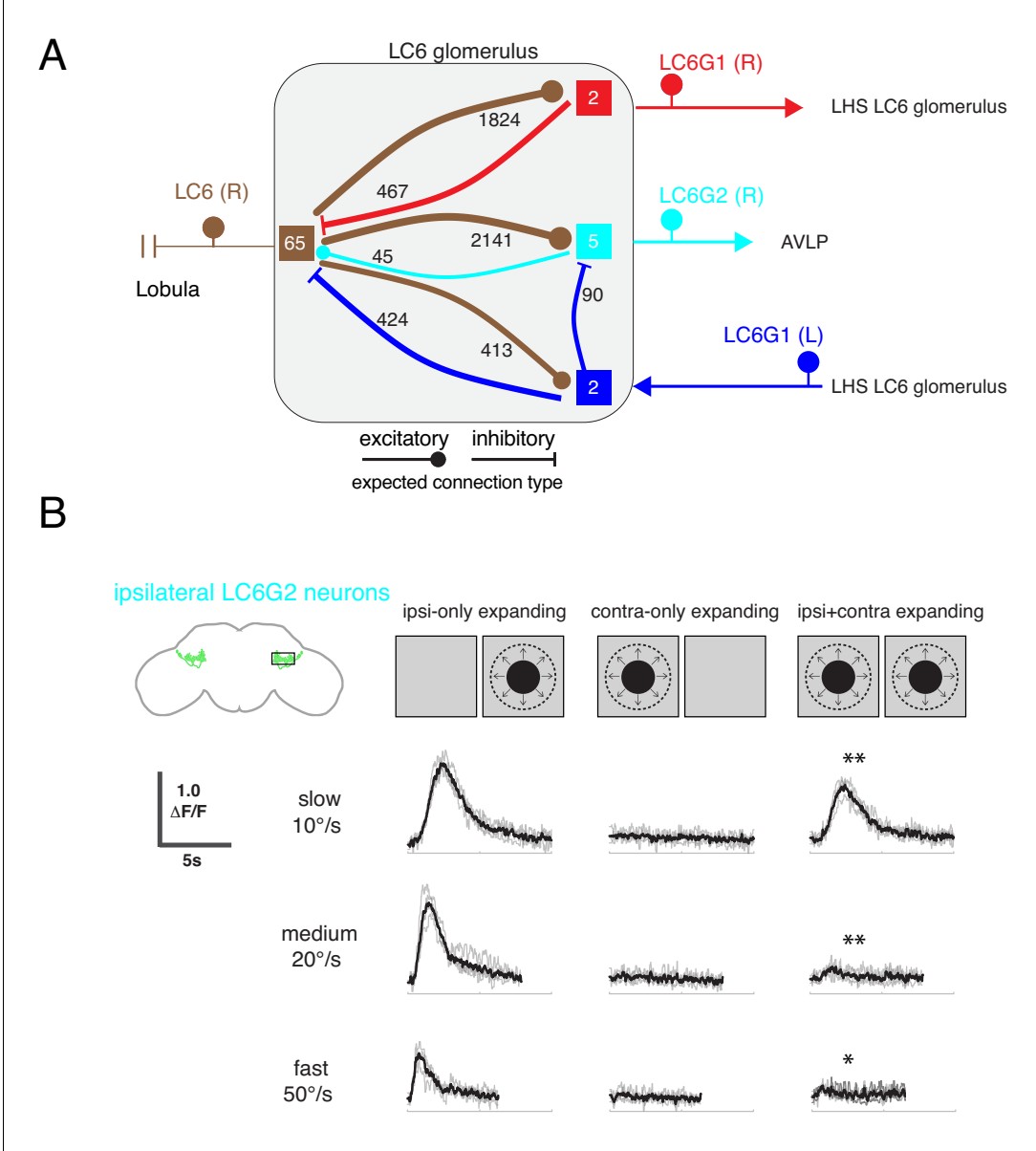

**Figure 8.** Interconnections between the LC6 glomeruli support contralateral suppression of visual responses to looming stimuli. (**A**) A proposed microcircuit between LC6s, LC6G1, and LC6G2. Synapse counts are shown for observed connections in the EM reconstruction (above threshold of 15 synapses). Each colored box indicates one of the four groups of neurons in the connectivity matrix (*Figure 7B*), and the numbers of cells within each group are listed within each box. Based on expression of neurotransmitter markers (*Figure 1—figure supplement 3*), LC6 and LC6G2 are expected to be excitatory (indicated by the ball-shaped terminal), and LC6G1s are expected to be inhibitory (bar-shaped terminal). Within the glomerulus the line thickness indicates the number of synapses in each connection type (logarithmically weighted), which is also noted next to each connection. Outside of the glomerulus, arrowheads are used to depict the direction of the projection, away or towards the glomerulus. (**B**) Evoked calcium responses in LC6G2 by variations of bilateral looming stimuli. Looming stimuli are centered at ±45˚. Responses for three different, constant speeds of looming. Mean responses from individual flies are in gray and mean responses across flies in black (N=4). The LC6G2 response is significantly reduced by the simultaneous presentation of a contralateral looming stimulus at all speeds (p = 2.2 × 10$^{-3}$, p = 4.8 × 10$^{-3}$, p = 2.3 × 10$^{-2}$ at 10, 20, 50˚/s respectively, two-sided paired t-test). See *Supplementary file 2* for summary of all looming visual stimuli presented.

The online version of this article includes the following figure supplement(s) for figure 8:

**Figure supplement 1.** Hemibrain connectivity.

*Figure 8 continued on next page*

*Figure 8 continued*

**Figure supplement 2.** LC-LC synapses in the hemibrain dataset.

ipsilateral only stimulus condition; *Figure 8B*, column 3). This result agrees well with our working model in which the responses of the LC6G2 glomerulus projection neurons are suppressed by the presence of the preferred (looming) stimulus from the contralateral eye, mediated by inhibition from the bilaterally projecting LC6G1 neurons. We have confirmed all of these connections in a newer EM dataset (*Scheffer et al., 2020*) (see discussion – section 'Assessing major results in light of new fly central brain connectomic data' and *Figure 8—figure supplements 1* and *2*), further supporting the anatomical basis of the simple model of *Figure 8A*. Together these results suggest that contralateral visual inputs further enhance the spatial selectivity of these LC6 projection neurons.

## Discussion

In this study, we investigated the circuitry downstream of the looming responsive LC6 visual projection neurons. We used light-level anatomical analysis to identify candidate downstream neurons (*Figure 1* and *Figure 1—figure supplement 1*) and then used intersectional genetic methods to establish driver lines that target each of these putative downstream target neurons (*Figure 1—figure supplement 2*). We used these driver lines to show that five of these cell types are functionally connected to LC6 neurons (*Figure 2*). We then selected two of these cell types, the bilaterally projecting LC6G1 neurons and the ipsilateral LC6G2 projection neurons for further functional studies. We found that these two cell types differentially encode looming stimuli (*Figures 3* and *4*). The LC6G1 neurons responded to looming stimuli over a large part of the visual field and showed nearly identical stimulus selectivity as the LC6 neurons. By contrast, the LC6G2 neurons responded to looming stimuli within a more restricted region of the visual field, and showed enhanced stimulus selectivity, preferentially encoding looming stimuli, while being less responsive to non-looming stimuli than their LC6 inputs.

The organization of visual projection neuron axons into glomeruli in the fly brain has been the subject of significant speculation. One popular proposal, inspired by the apparent loss of retinotopy within the glomeruli, is that these structures represent a transformation from a visual, spatial 'where' signal to a more abstract 'what' representation (*Mu et al., 2012*; *Strausfeld et al., 2007*; *Wu et al., 2016*). However, we found that the LC6 glomerulus target neurons are spatially selective (*Figure 3*), and so we pursued a connectomic reconstruction of the LC6 glomerulus and showed that while some retinotopic information is present in the LC6 glomerulus (*Figures 5* and *6*), the spatial receptive fields of the LC6G neurons appear to be established by biased synaptic connectivity (*Figure 7*), that cannot be explained by simple innervation of the glomerulus (*Figure 7—figure supplement 1*). Finally, our connectomics analysis clarifies the flow of information between these target neurons and LC6 (*Figure 8*).

From this functional and anatomical evaluation of the circuit, the picture that emerges is of glutamatergic (putative inhibitory) LC6G1 neurons that serve as interneurons within and between the glomeruli, reporting the summed LC6 activity, which is used to suppresses high (or broad) levels of activation—as would be encountered during forward locomotion in a cluttered environment. In contrast, the cholinergic (excitatory) LC6G2 projection neurons, convey the detection of looming stimuli to the AVLP, a deeper brain area. These neurons show enhanced selectivity for looming stimuli within one eye (*Figure 4*), but also receive inhibitory inputs (at least in part mediated via LC6G1 neurons, *Figure 8*) that would further serve to preserve responses to looming stimuli while suppressing responses to global visual stimulation.

### Organization of visual space in the LC6 glomerulus

We have investigated how visual-spatial information, representing different retinotopic coordinates, is organized in the LC6 glomerulus through light microscopy and EM anatomy. From confocal images, the LC6 axons (*Figure 1C'*) do not show any retinotopic organization along either the long or short axes of the glomerulus. However, from our EM reconstruction of the LC6 glomerulus (*Figures 5*, *6* and *7*), we found spatial organization at multiple levels. (1) In following the LC6 axons into

the central brain, the retinotopic organization of the lobula is not apparent along the axon tract but is found at a modest level in the glomerulus (*Figure 5I*). (2) We found an approximate correspondence between the tip position of each LC6 axon in the glomerulus and the visual coordinates of that neuron (*Figure 6C,D*), and accordingly we find that the LC6 synapses in some compartments along the long axis of the glomerulus exhibit a visual-spatial bias (*Figure 6—figure supplement 1B, C*). (3) The LC6 pre-synapses show modest retinotopic organization at the finest scale we examined (*Figure 6I*) and the axo-axonic connections between LC6 neurons in the glomerulus are biased such that visually nearby neurons are more strongly connected (*Figure 6—figure supplement 1A*). (4) The LC6 downstream neurons receive spatially biased connections from LC6 neurons in the glomerulus (*Figure 7*). The synapses of LC6G1 and G2 were distributed broadly across the LC6 glomerulus (*Figure 7—figure supplement 3*); however, by forming higher numbers of synapses onto subsets of the LC6 neurons throughout the glomerulus, these target neurons access inputs corresponding to restricted regions of the visual field (*Figure 7* and *Figure 7—figure supplements 1* and *2*). Taken together, we find stronger evidence for non-random spatial organization that can conveys retinotopic information in the LC6 glomerulus at the EM level than we expected based on light microscopy (*Figure 1*). However, none of the features of the thus-far-described retinotopic organization appears to explain the EM-estimated and functionally measured receptive fields of the LC6G neurons (*Figure 7—figure supplement 2*). Instead we find that the largely overlapping processes of individual LC6G neurons (especially the LC6G2 neurons), through fine-scale selective connectivity, integrate visual-spatial information from restricted eye regions. This situation is somewhat analogous to the well-studied T4 and T5 neurons, where the dendrites of these neurons, within retinotopically organized neuropils, prominently overlap with many cells of the same and different subtypes, and yet establish precise connections to specific cell types only at specific dendritic locations (*Shinomiya et al., 2019b*). While visual-spatial readout is supported by connectivity in the glomerulus, we find strong evidence for a coarsening of the visual representation, as neurons exhibit increasingly larger field of views, with all target neurons integrating from the majority of LC6 neurons. Further reconstructions and analysis will be required to determine if these transformations described in the LC6 glomerulus are a general circuit strategy employed within the other optic glomeruli.

What benefits, if any, might this peculiar glomerulus structure serve for the organization of visual projection neurons? One prevailing view is that retinotopy affords wiring efficiency for downstream cells, provided they benefit from reading out information from neighboring visual regions (*Chklovskii and Koulakov, 2004*). On the other extreme, if the readout strategy requires non-spatial, random access to the entire visual field, then a compact, ball-like structure would minimize wire length. But if the purpose of some downstream neurons is to access all of the inputs (e.g. LC6G1) while others integrate from a smaller region of the visual field (e.g. LC6G2), then the organization we observe in the LC6 glomerulus presents a sensible compromise—a compact structure which minimizes overall wire length for 'global integrating' neurons, while still allowing retinotopic readout, albeit somewhat coarsened, to support spatial vision.

## Limitations of comparisons between the functional and anatomical receptive fields of LC6 downstream cells

We have measured the receptive field of LC6 downstream cells LC6G1 and LC6G2 using two very different methods, functional in vivo $Ca^{2+}$ imaging and estimates based on mapping EM-reconstructed neurons and their connections. We find some overlap between the EM-predicted receptive fields and the functionally measured ones, for each cell type, especially in the degree to which both measures show LC6G1 RF support near the midline and LC6G2 support away from the midline. While we took great care to align the data sets so as to produce an accurate comparison between the RF measurements, there are several substantial challenges that likely limited the concordance in the direct comparison of the RF maps we present in *Figure 7*. One limitation is that our functional RF measurements, which we considered to be very comprehensive at the time we designed those experiments, only sampled a rather limited field of view, while the EM-estimated RFs nearly cover one hemisphere per eye. We believe that in future experiments, it will be important to sample responses across as much of the eye as possible, which should provide a much stronger basis for anatomical-functional comparisons.

The anatomical RF maps were estimated by only taking into account the feedforward LC6 inputs, the direct synapses from LC6s onto LC6G neurons. Connections within the glomerulus, from either

excitatory or inhibitory neurons with their own visual-spatial biases, would be expected to reweight the resultant RFs (by either broadening or sharpening the spatial tuning). The anatomical RFs were made by applying equal weights for all of the LC6 synapses. This assumption likely explains why the EM-estimated LC6G RFs exhibit peaks around where the LC6 density is highest, a feature not reflected in the functional measurements (these peaks could be flattened by either lower weights in regions of higher density or by the effect of inhibitory interneurons in the glomerulus). Connections outside of the LC6 glomerulus, which are not included in the EM analysis, could also contribute to the functionally measured LC6G RFs. In the case of LC6G2, these neurons also receive feedforward input from LC16s (*Figure 1* and discussion section 'Assessing major results in light of new fly central brain connectomic data'), which are also sensitive to looming visual motion (*Wu et al., 2016*). An exciting follow-up analysis would examine how well the spatial bias in connectivity from one glomerulus is matched with the spatial bias of inputs in another glomerulus. Accounting for these network effects, which undoubtedly contribute to LC6G RFs, is currently impractical. Many of these additional connections are unknown, it is unclear how to weight them, and even less clear how to constrain the dynamics of these contributions during the course of the looming stimulus response.

Fundamentally, we lack a principled method for determining how to weight the contribution of each synapse, which is especially unclear when comparing between excitatory and inhibitory synapses on an axon terminal. Comparing our functional connectivity and EM connectivity data (*Figure 3*), it does not appear that LC6 downstream activity scales with number of LC6 synaptic inputs. For the two cell types traced in the EM volume (LC6G1 and LC6G2), the difference in the number of synaptic inputs from LC6 neurons is rather small (~2-fold, per cell), whereas the difference in the calcium response (to the same LC6 stimulation) was almost 40-fold. This difference may be attributed to a large number of issues, including the network effects described above. From the EM data, we also found that connectivity between LC6, LC6G1, and LC6G2 is bidirectional (*Figure 7B*), which further complicates this picture.

Another difference between the EM-estimated maps and the functional RF maps is that the functional measurements represent an average of multiple cells in multiple animals, whereas the EM maps used single cells from one hemisphere of one animal. The RF seems to be slightly variable between animals (*Figure 3—figure supplement 1*), such that individual differences could account for part of the discrepancy we observe between the maps.

Given these myriad differences, it is rather surprising that these maps should agree at all. And yet the anatomical RF maps indicate that an equal weighting of LC6 inputs alone is sufficient to provide differential spatial information to these downstream cell types. However, since the anatomical RF maps only partially agree with their functional RF map counterparts, we expect that network effects including from neurons not considered in our analysis, non-uniform synaptic weighting, and fly-to-fly variability, may all contribute to the resultant RFs of the LC6G neurons.

## Assessing major results in light of new fly central brain connectomic data

The recent availability of a dense EM reconstruction of a large portion of the fly brain (*Scheffer et al., 2020*), including the optic glomeruli and part of the lobula, allowed us to revisit our connectomics results in a second EM volume. This 'hemibrain' dataset is thought to provide close to complete coverage of neurons and strong synaptic connections within the reconstructed volume, making it well-suited to validate and extend our analyses of LC6 synaptic partners. However, in view of a potential limitation of the hemibrain dataset for analyses that rely on precise synapse counts—many synapses remain unassigned to specific neurons (these synapses are associated with small fragments that have not been merged into a larger reconstructed cell)—we did not attempt to repeat our work on the representation of visual-spatial information in the LC6 glomerulus in this volume, as this would require extensive proofreading of LC6 synapses.

To identify LC6G neurons in the hemibrain, we compared the anatomy of EM reconstructions to light microscopy (LM) images (see *Figure 1*) of LC6G cells. For each LC6G type, we identified two to nine hemibrain counterparts (for details, including some limitations of the matching process, see Material and methods, section 'EM analyses: hemibrain data' and illustrations in *Figure 8—figure supplement 1A*; hemibrain identifiers of cells annotated as LC6Gs are in *Supplementary file 4*). By examining synaptic connections of these cells in the hemibrain, we found that all LC6G cell types are direct postsynaptic partners of LC6 (*Figure 8—figure supplement 1B*), confirming our EM

findings for LC6G1 and LC6G2 and the predictions from light microscopy and functional connectivity for LC6G3, LCG4 and LC6G5 (though a few putative LC6G5 reconstructions get only minor or no LC6 input).

The hemibrain data indicate that LC6G neurons are far from the only postsynaptic partners of LC6 cells (of a total of 35 cells that are postsynaptic to the group of LC6s at more than 100 synapses, i.e. an average of ~2 connections per LC6 cell, only 18 are matched to the 5 LC6G cell types). However, as types, the 5 LC6G neurons are among the strongest LC6 targets, as judged by the total number of synaptic connections from LC6 cells (*Figure 8—figure supplement 1B*; data for individual cells, not aggregated by type, are included in *Supplementary file 4*). In addition, while all LC6G cells receive input from cell types other than LC6, including in some cases from other LCs (*Figure 8— figure supplement 1B*, *Supplementary file 4*), LC6G cells are among the cells with the highest relative LC6 input: Of the 35 LC6 target cells with at least 100 input synapses from LC6, all 18 LC6G cells but only 3 of the 17 other LC6 targets receive at least 15% of their input synapses from LC6 cells. The high proportion of LC6 synapses among LC6G inputs suggests LC6 input is a major determinant of LC6G responses, in particular to looming stimuli. However, experimental approaches, such as imaging LC6G responses while blocking LC6 activity, will be required to better understand the integration of signals in the targets of LC6 or other LC neuron types. A notable example of LC6G cells that receive input from other LC types, primarily LC16, are LC6G2 neurons, consistent with our light microscopy result that LC6G2 arbors overlap with the LC16 glomerulus.

Our manual EM reconstructions indicate that LC6G1 cells are both major LC6 targets and inputs (*Figure 7B*). These findings are confirmed by the hemibrain data. In combination, LC6G1 and its contralateral counterpart (here identified as LC6G1L) have the largest number of synapses to (*Figure 8— figure supplement 1C*) and from (*Figure 8—figure supplement 1B*) LC6 compared to all other cell types, with only the population of LC6 cells themselves coming close. We note that none of the remaining strong LC6 inputs appear to connect LC6 cells in the left and right glomeruli directly. In addition, while there are additional cells with a morphology similar to LC6G1 that also synapse in the LC16 or LC6 glomeruli, none of these appear to be both strong LC6 inputs and outputs (as predicted from a combination of LM and EM data). This indicates that LC6G1 cells are the main, if not only, direct connection between left and right brain LC6 cells. Interestingly, a very similar cell type (identified as LC16G1 in *Supplementary file 4*) provides similar strong connections between LC16 cells in the two brain hemispheres. Another component of our circuit model in *Figure 8*, contralateral LC6G1 input to LC6G2, is also consistent with the hemibrain data (not shown).

Apart from LC6G1, the main synaptic input to LC6 cells in the glomerulus is from other LC6 cells, both in our reconstructions and in the hemibrain dataset. We wanted to know whether this feature was specific to LC6 connectivity and therefore examined LC-LC synapses for 12 other LC types (corresponding to the largest PVLP glomeruli as well as LC26, which has a glomerulus close to LC6 and LC16 (*Figure 8—figure supplement 2A*)) All these cells also had many LC-LC connections, that account for a substantial percentage of inputs and outputs. Notably, the LC6-LC6 synaptic connections are seen to be neither unusually high nor low when compared to the 12 other cell types. Taken together, this extended analysis of the hemibrain dataset provides an independent confirmation of the major results about cell identity and connectivity that we reported based on the more targeted, manual EM reconstructions.

## Spatial detection of visual looming

Given that LC6 downstream cells respond to looming stimuli (*Figure 4*), how might the organization of visual-spatial information in the LC6 glomerulus contribute to the fly's ability to detect looming features? At the level of LC6 population, the neighboring LC6-LC6 connections might contribute to the detection of looming. We only described axo-axonic connections, but dendro-dendritic connections between overlapping neurons are also expected. While we now know that axo-axonic connections are a common feature of LC neurons (*Figure 8—figure supplement 2*), it is also likely that the lateral facilitation of LC6 neurons could enhance responses to looming stimuli by this population of non-directionally selective cells. The first cells that are excited by a small dark object would facilitate the neighboring cells, and as the object grows in size, the surround facilitation would amplify responses while spreading as an outwardly radiating wave. At the same time, this scenario would not enhance responses to a receding object, which is consistent with the LC6 population responses (*Figure 3*; *Wu et al., 2016*). At the level of LC6 downstream cells, we have shown that LC6G1 and

LC6G2 display different levels of selectivity for looming features (*Figure 4*). While determining the mechanism of this enhanced selectivity for looming is beyond the scope of this paper, the observation of this selectivity enhancement strongly suggests that the LC6 pathway is specialized for the detection of looming stimuli, since targets of this pathways are even more selective for looming stimuli than the visual projection neurons.

Since the LC6G1 and LC6G2 cells exhibit different receptive fields (*Figure 3*), it is likely that the downstream cell types act as different filters for particular looming features in particular regions of space. The kind of spatial and stimulus selectivity we measured for LC6G2 neurons (*Figures 3* and *4*) could support directional escapes, where approaching stimuli elicit escape take-offs in roughly the opposite direction (*Card and Dickinson, 2008*). Further, it seems essential for the fly to discriminate between the ground looming upwards in the ventral visual field and an approaching object in the medial visual field. These events would require triggering significantly different motor programs, such as landing or evasion. Therefore, the organization we observe in LC6 target neurons (especially LC6G2) might provide an efficient circuit logic to transform a purely sensory input into a signal that drives specific motor reactions, such as directed escapes. From this perspective, the coarsened visual-spatial readout in the LC6 glomeruli may be the expected feature of an intermediate step in a sensory-to-motor transformation.

## The role of contralateral suppression

Through our EM reconstruction, we found interconnectivity between LC6, LC6G1, and LC6G2 within the glomerulus. Together with their neurotransmitter identities (*Figure 1—figure supplement 3*), we proposed a simple circuit within the LC6 glomerulus capable of contralateral suppression (*Figure 8*), which we further confirmed is supported by the newer central brain connectome data (see discussion above). LC6G1 neurons, that we showed are glutamatergic, participate in recurrent loop (directly via the LC6 neurons) between the LC6 glomeruli. So while glutamate can act as both an excitatory and inhibitory transmitter in the fly CNS (depending on the receptors expressed by the post-synaptic neurons), because of this reciprocal connectivity across the midline, we think it is far more likely that the net effect of the LC6G1 connection is to be inhibitory. We tested this circuit hypothesis using bilateral looming stimuli while imaging from LC6G2 and found that the stimulus-evoked responses were consistent with the proposed circuit mechanism. Contralateral suppression was already implied from light level anatomy that showed the bilateral downstream cell type specifically contacted the same glomeruli in both hemispheres. This type of connection between glomeruli is common; based on our analysis of Janelia's GAL4 collection, we expect that most if not all optic glomeruli are connected by one or more bilaterally projecting interneuron types. For example, we confirm, using the hemibrain data, that the LC16 cells are connected by neurons very similar to LC6G1. Contralateral suppression has been observed in other contexts and brain regions in *Drosophila*, such as detection of visual bilateral bar stimuli (*Sun et al., 2017*) or wind direction (*Suver et al., 2019*), in the latter case being implemented by bilaterally projecting cell types of similar morphology to LC6G1. This suggests that the use of bilaterally projecting cell types might be a common strategy used for contralateral suppression in the *Drosophila* CNS. In addition to being common, it is also sensible (and is related to the engineering principle of designing amplifiers with 'common mode rejection'). A major goal of sensory systems appears to be the detection and localization of specific sensory cues, and yet in the natural world cues are rarely discrete, they emerge in complex mixtures across space and time. From this perspective, one of the simplest methods to identify a specific, localizable threat, such as an approaching object, is to compare whether this detection event is restricted to one side of the animal, or whether it is prominent on both sides. We propose that contralateral suppression is likely to be a basic strategy used in establishing stimulus selectivity.

## Outlook

Here, we described multiple levels of spatial organization within the LC6 glomerulus synthesizing data from in vivo calcium imaging, functional connectivity, and light and EM level anatomy. Using celltype-specific driver lines (*Wu et al., 2016*) and the recently completed full adult fly brain EM volume (*Zheng et al., 2018*), we correlated neurophysiological measurements with EM connectivity between identified cell types. As a result, we have described how visual-spatial information is read

out from a neuropil with limited retinotopic organization and proposed a circuit between LC6 and LC6 downstream cells that enables contralateral suppression. As EM-level connectomes become available, the approach we used of integrating functional analysis with circuit mapping should greatly accelerate the discovery of mechanisms for diverse circuit computations.

# Materials and methods

## Key resources table

| Reagent type (species) or resource | Designation | Source or reference | Identifiers | Additional information |
|---|---|---|---|---|
| Genetic reagent (*Drosophila melanogaster*) | OL0070B | https://doi.org/10.7554/eLife.21022.001 | OL0070B | LC6 split-GAL4 driver line available via https://www.janelia.org/split-GAL4. |
| Genetic reagent (*D. melanogaster*) | OL0081B | https://doi.org/10.7554/eLife.21022.001 | OL0081B | LC6 split-GAL4 driver line available via https://www.janelia.org/split-GAL4. |
| Genetic reagent (*D. melanogaster*) | SS00825 | This paper | SS00825 | LG6G1 split-GAL4 driver line, available via https://www.janelia.org/split-GAL4. |
| Genetic reagent (*D. melanogaster*) | SS00824 | This paper | SS00824 | LG6G1 split-GAL4 driver line, available via https://www.janelia.org/split-GAL4 |
| Genetic reagent (*D. melanogaster*) | SS02036 | This paper | SS02036 | LG6G2 split-GAL4 driver line, available via https://www.janelia.org/split-GAL4 |
| Genetic reagent (*D. melanogaster*) | SS02099 | This paper | SS02099 | LG6G3 split-GAL4 driver line, available via https://www.janelia.org/split-GAL4 |
| Genetic reagent (*D. melanogaster*) | SS02699 | This paper | SS02699 | LG6G3 split-GAL4 driver line, available via https://www.janelia.org/split-GAL4 |
| Genetic reagent (*D. melanogaster*) | SS03690 | This paper | SS03690 | LG6G4 split-GAL4 driver line, available via https://www.janelia.org/split-GAL4 |
| Genetic reagent (*D. melanogaster*) | SS02410 | This paper | SS02410 | LG6G5 split-GAL4 driver line, available via https://www.janelia.org/split-GAL4 |
| Genetic reagent (*D. melanogaster*) | SS02409 | This paper | SS02409 | LG6G5 split-GAL4 driver line, available via https://www.janelia.org/split-GAL4 |
| Genetic reagent (*D. melanogaster*) | SS25111 | This paper | SS25111 | LG6G5 split-GAL4 driver line, available via https://www.janelia.org/split-GAL4 |
| Genetic reagent (*D. melanogaster*) | SS03641 | This paper | SS03641 | LG26G1 split-GAL4 driver line, available via https://www.janelia.org/split-GAL4 |

*Continued on next page*

*Continued*

| Reagent type (species) or resource | Designation | Source or reference | Identifiers | Additional information |
|---|---|---|---|---|
| Genetic reagent (*D. melanogaster*) | pJFRC51-3XUAS-IVS-Syt::smHA in su(Hw)attP1, pJFRC225-5XUAS-IVS-myr::smFLAG in VK00005 | https://doi.org/10.1073/pnas.1506763112 | | Presynaptic and membrane marker |
| Genetic reagent (*D. melanogaster*) | MCFO-1: pBPhsFlp2::PEST in attP3;; pJFRC201-10XUAS-FRT > STOP > FRT-myr::smGFP-HA in VK0005, pJFRC240-10X-UAS-FRT>STOP > FRT-myr::smGFP-V5-THS-10XUAS-FRT > STOP > FRT-myr::smGFP-FLAG in su(Hw)attP1 | https://doi.org/10.1073/pnas.1506763112 | RRID:BDSC_64085 | MultiColor FlipOut (stochastic labeling) |
| Genetic reagent (*D. melanogaster*) | 42E06- LexA (JK22C) | https://doi.org/10.1534/genetics.114.173187 | | |
| Genetic reagent (*D. melanogaster*) | 13XLexAop2-CsChrimson-tdT (attP18), 20XUAS-IVS-Syn21-opGCaMP6f p10 (Su(Hw)attp8) | https://elifesciences.org/articles/49257/ and this paper | | Functional connectivity |
| Genetic reagent (*D. melanogaster*) | pJFRC7-20XUAS-IVS-GCaMP6m (VK00005) | 10.1038/s41592-019-0435-6 | RRID:BDSC_52869 | |
| Genetic reagent (*D. melanogaster*) | pJFRC48-13XLexAop2-IVS-myrtdTomato (su(Hw)attP8) | This paper | | |
| Genetic reagent (*D. melanogaster*) | UAS-7xHaloTag::CAAX (VK00005) | https://doi.org/10.1534/genetics.116.199281 | | HaloTag reporter used for FISH |
| Sequence-based reagent | FISH probes for GAD1, VGlut and ChaT | https://doi.org/10.1038/nmeth.4309 | | |
| Antibody | Anti-HA rabbit monoclonal C29F4 | Cell Signaling Technologies: 3724S | RRID:AB_1549585 | (1:300) |
| Antibody | Anti-FLAG rat monoclonal DYKDDDDK Epitope Tag Antibody [L5] | Novus Biologicals: NBP1-06712 | RRID:AB_1625981 | (1:200) |
| Antibody | DyLight 550 conjugated anti-V5 mouse monoclonal | AbD Serotec: MCA1360D550GA | RRID:AB_2687576 | (1:500) |
| Antibody | DyLight 549 conjugated anti-V5 mouse monoclonal | AbD Serotec: MCA1360D549GA | RRID:AB_10850329 | (1:500) |
| Antibody | Anti-Brp mouse monoclonal nc82 | DSHB | RRID:AB_2314866 | (1:30) |
| Software, algorithm | Python 3.7.8 | http://www.python.org | RRID:SCR_008394 | |
| Software, algorithm | neuprint python | https://github.com/connectome-neuprint/neuprint-python | | |

*Continued on next page*

*Continued*

| Reagent type (species) or resource | Designation | Source or reference | Identifiers | Additional information |
|---|---|---|---|---|
| Software, algorithm | FluoRender | http://www.sci.utah.edu/software/fluorender.html | RRID:SCR_014303 | |
| Software, algorithm | VVD_Viewer | https://github.com/takashi310/VVD_Viewer | | |
| Software, algorithm | Fiji | http://fiji.sc | RRID:SCR_002285 | |
| Software, algorithm | R | https://www.r-project.org/ | | Anatomical analysis |
| Software, algorithm | R studio | http://www.rstudio.com/ | | |
| Software, algorithm | Natverse package in R | http://natverse.org | | EM anatomy tools |
| Software, algorithm | MATLAB | Mathworks, Inc | Multiple versions 2016–2019 | Visual stimuli, experiment control, data analysis, and eye alignment |
| Software, algorithm | Nia (neuron image analysis) | https://bitbucket.org/jastrother/neuron_image_analysis | | |
| Software, algorithm | LED Panel display tools | https://github.com/reiserlab/Modular-LED-Display (Generation 3) | | |
| Software, algorithm | Thunder-project | https://github.com/thunder-project/thunder | | http://thunder-project.org/ image registration – functional connectivity |

## Fly lines: split-GAL4 generation and anatomical analyses (light microscopy level)

To construct split-GAL4 lines for potential LC6 target neurons, we first selected candidate AD and DBD hemi drivers by visually searching images of GAL4 driver expression patterns ( *Jenett et al., 2012*; *Tirian and Dickson, 2017*). Typically, we tested several candidate split-GAL4 combinations for each target cell type. The AD and DBD hemi drivers we used are from published collections (*Dionne et al., 2018*; *Tirian and Dickson, 2017*). A detailed description of split-GAL4 hemi drivers is also available online (https://bdsc.indiana.edu/stocks/gal4/split_intro.html). The cell-type-specific split-GAL4 lines generated in this study together with original images of their anatomy are available at the Janelia split-GAL4 website (https://splitgal.janelia.org/cgi-bin/splitgal4.cgi), under 'LC6 downstream 2020'. All fly lines used throughout the manuscript are summarized in *Supplementary file 1*.

To visualize the overall expression patterns of the split-GAL4 driver lines, we used previously described reporter transgenes (pJFRC51-3XUAS-IVS-Syt::smHA in su(Hw)attP1 and pJFRC225-5XUAS-IVS-myr::smFLAG in VK00005 [*Nern et al., 2015*] and methods (*Aso et al., 2014*; *Wu et al., 2016*). Detailed protocols are also available online (https://www.janelia.org/project-team/flylight/protocols) under 'IHC - Anti-GFP', 'IHC - Polarity Sequential' and 'DPX mounting'). For stochastic labeling of individual cells, we used standard protocols for MultiColor FlpOut (MCFO) (*Nern et al., 2015*). Briefly, split-GAL4 lines were crossed to MCFO-1 and adult progeny heat-shocked and immunolabeled with antibodies against FLAG, HA and V5 epitopes as described (*Nern et al., 2015*). Detailed protocols are also available online (https://www.janelia.org/project-team/flylight/protocols) under 'IHC - MCFO'. We examined expression of the neurotransmitter markers GAD1, VGlut, and ChaT by Fluorescence in situ Hybridization (FISH) using published protocols and probe sets

(*Long et al., 2017*; *Meissner et al., 2019*). We note that in general we used the same driver lines across our experiments, with one noteworthy exception. We developed a more specific driver line for LC6G5 (SS25111) in the later stages of this project. This line was used only for the FISH experiments (*Figure 1—figure supplement 3*, *Supplementary file 1*), where interpretation of results benefits from the cleanest available driver line.

Images were acquired on Zeiss LSM 710 or 800 confocal microscopes with 20 × 0.8 NA or 63 × 1.4 NA objectives. We used Fiji (http://fiji.sc) to generate maximum intensity projections of driver patterns and sections of FISH images. For these images, adjustments were limited to changes of brightness and contrast. Most other anatomy panels show composites of multiple registered images, which were generated using a recently described template brain (*Bogovic et al., 2018*). To more clearly display the cells of interest images used in these panels were, in some cases, manually edited to exclude additional cells or background present in the original images. Editing and assembly of these composites was mainly done using FluoRender (http://www.sci.utah.edu/software/fluorender.html) or the related VVD_Viewer (https://github.com/takashi310/VVD_Viewer).

## Fly lines: functional connectivity

For the functional connectivity experiments (*Figure 2*), in order to express Chrimson (*Klapoetke et al., 2014*) and GCaMP6f (*Chen et al., 2013*) in different neurons, LexA/LexAop and split-GAL4/UAS systems were simultaneously used in the same animal. For exploring specific downstream candidate neurons, a LexA line containing the LC6 pattern (R42E06 in JK73A, *Knapp et al., 2015*) was used to drive Chrimson in LC6, while the split-GAL4 lines (SS0825, SS2036, SS2099, SS3690, SS2409, SS3641) were used to drive expression of opGCaMP6f (codon-optimized GCaMP6f) in candidate LC6 downstream neurons (summarized in *Supplementary file 1*). Flies were reared under standard conditions (60% humidity, 25°C) on a cornmeal agar diet supplemented with retinal (all-trans-retinal 0.2 mM concentration in the food) in vials that were wrapped in foil to keep flies in the dark to prevent spurious activation of Chrimson by ambient light. Flies were collected following eclosion and held under the same rearing conditions until experiments were performed.

## Fly lines: calcium imaging

Cell-type-specific expression of the fluorescent calcium indicator GCaMP6m (*Chen et al., 2013*) was achieved using the split-GAL4/UAS expression system (*Luan et al., 2006*; *Pfeiffer et al., 2010*). The GAL4 driver lines were constructed using methods described above. All flies used for calcium imaging experiments were reared under standard conditions (25°C, 60% humidity, 12 hr light/12 hr dark, standard cornmeal/molasses food), and all imaging experiments were performed on females 3–6 days post-eclosion. To image from LC6 and its downstream targets, split-GAL4 driver lines [LC6: OL0070B, LC6G1(bilaterally projecting downstream): SS0825, LC6G2 (ipsilaterally projecting downstream): SS2036] were crossed to pJFRC7-20XUAS-IVS-GCaMP6m in VK00005 (DL background) effector line. In a subset of the recordings the LC6 neurons, and thus the glomerulus, were labeled with tdTomato (to aid in selecting appropriate imaging plane). The lines used in each experimental Figure panel are summarized in *Supplementary file 1*.

## Functional connectivity: preparation, experimental details, and data analysis

Brains from female adult flies 1–3 days post-eclosion, were isolated by dissecting the head in a saline bath (103 mM NaCl, 3 mM KCl, 2 mM CaCl$_2$, 4 mM MgCl$_2$, 26 mM NaHCO$_3$, 1 mM NaH$_2$PO$_4$, 8 mM trehalose, 10 mM glucose, 5 mM TES, bubbled with 95% O$_2$/5% CO$_2$). The brain was then placed on a poly-lysine-coated coverslip (neuVitro, Vancouver, WA, GG-12-PDL) posterior side up and perfused with saline (same composition as above, 21°C) (*Figure 3A*). Images of the brain were acquired using a two-photon microscope (Custom made at Janelia by Dan Flickinger and colleagues, with a Nikon Apo LWD 25 × NA1.1 water immersion objective #MRD77225). The standard imaging mode was a 512 × 512 image at 2.5 frames/s, and a ~ 353 µm ×~353 µm field of view (~0.69 µm ×~0.69 µm / pixel). The sample was imaged using a near-infrared laser (920 nm, Spectra Physics, Insight DeepSee) that produced minimal collateral activation of Chrimson at our typical imaging power.

The light-gated ion channel Chrimson was activated by 590 nm light (Thorlabs M590L3-C1) presented through the objective. Photoactivation light was delivered in a pulse train that consisted of

six 1 s pulses (within each 1 s pulse: square-wave modulation at 50 Hz, 10% duty cycle, 30 s inter-pulse interval). Two stimulation protocols, in succession, were used on each brain, the first (*Figure 2*) in which the light intensity increased for each of the six pulses (0.12, 0.24, 0.48, 0.72, 0.96, 1.21 mW/mm$^2$; measured using Thorlabs S170C microscope slide light power meter) and the second (*Figure 2—figure supplement 1*) in which the light intensity was kept constant for each of the six pulses (0.48 mW). Stimulation light was spatially modulated using a DMD (Digital Micromirror Device, Texas Instruments, DLP LightCrafter v2.0), and was restricted to one or both LC6 glomeruli depending on the experiment (*Figure 2*: left glomerulus only stimulation and ROI on left glomerulus region; *Figure 2—figure supplement 1* left and right glomerulus only stimulation and ROI on left glomerulus region). This spatial restriction limited the activation of other few non-specific cells labeled with the same driver line.

Image registration was conducted using code from the Thunder package (https://github.com/thunder-project/thunder; *Freeman et al., 2014*).

The calcium responses of candidate LC6 downstream neurons to photoactivation were measured by calculating the ΔF/F for a manually drawn region of interest (ROI) in the imaging plane, which covered the largest section of its dendritic arborization. The ΔF/F was taken as (F-F0)/F0 where F is the instantaneous mean fluorescence of the ROI and F0 is the baseline fluorescence of the ROI. The baseline fluorescence was taken as the 10th percentile of the stimulation protocol period. Peak responses from each fly for both protocols were taken, and the mean ± SEM is shown as a stimulus response curve in *Figure 2* and *Figure 2—figure supplement 1*.

## In vivo two-photon calcium imaging: preparation

The imaging preparation was nearly identical to what we have described in a recent publication (*Strother et al., 2014*). Briefly, flies (mostly female, but some male) were cold anesthetized and tethered to a fine wire at the thorax using UV-curing adhesive. The two most anterior legs (T1) were severed and glued down along with the proboscis to prevent grooming of the eyes and to immobilize the head. Tethered flies were glued by the head capsule into a fly holder and after addition of saline (103 mM NaCl, 3 mM KCl, 1.5 mM CaCl$_2$, 4 mM MgCl$_2$, 26 mM NaHCO$_3$, 1 mM NaH$_2$PO$_4$, 8 mM trehalose, 10 mM glucose, 5 mM TES, bubbled with 95% O$_2$/5% CO$_2$; final Osm = 283, pH = 7.3; modified from *Wilson and Laurent, 2005*) to the bath, the cuticle at the back of the head was dissected away to expose the brain. Muscles 1 and 16 (*Demerec, 1965*) were severed to reduce motion of the brain within the head capsule, and the post-ocular air sac on the imaged side of the brain was removed to expose the optic glomeruli. The right side of the brain was always imaged (unless otherwise stated).

## In vivo two-photon calcium imaging: microscopy

The optic glomeruli were imaged using a two-photon microscope (Bruker/Prairie Ultima IV) with near-infrared excitation (930 nm, Coherent Chameleon Ultra II) and a 60x objective (Nikon CFI APO 60XW). The excitation power was 10–20 mW at the sample. The long excitation wavelength is not visible to the fly. Imaging parameters varied slightly between experiments but were within a small range around our typical acquisition parameters: 128 × 90 pixel resolution, and ~10 Hz frame rate (10.0–10.5 Hz). LC cell axon calcium data were collected from single planes selected to capture a consistently large slice of each glomerulus. In several experiments, the glomerulus was labeled with tdTomato and located with a separate red-channel detector. The filter sets used in the detection path were HQ525/70 bandpass (Chroma) and 510/42 bandpass (Semrock) for the green-channel, HQ607/45 bandpass (Chroma) and 650/54 bandpass (Semrock) for the red-channel.

## In vivo two-photon calcium imaging: visual stimulation

Visual stimulation details closely follow the methods used in our previous study (*Wu et al., 2016*). Flies were placed near the center of a modular LED display (*Reiser and Dickinson, 2008*) on which visual stimuli were presented (*Figure 3A*). The display consists of 574 nm peak output LEDs (Betlux ultra-green 8 × 8 LED matrices, #BL-M12A881UG-XX) covered with a gel filter (LEE #135 Deep Golden Amber) to greatly reduce stimulus emission at wavelengths that overlap with those of GCaMP emission, resulting in emission range of 560–600 nm (measured by Ocean Optics USB4000-UV-VIS spectrometer). The display was configured to cover 60% of a cylinder (32 mm ×32 mm

panels. forming 12 sides of an extruded 20-sided polygon, approximating a cylinder, each column is four panels high; total resolution is 32 pixels × 96 pixels). Further documentation about the (Generation 3) visual display is available at https://github.com/reiserlab/Modular-LED-Display.

Since the head is fixed into the fly holder with an approximately constant orientation (relative to horizontal, *Figure 3—figure supplement 1A*), the display was tilted to roughly match this angle. To estimate the projection of the display onto the flies' field of view, we first made careful measurements (angles and distances) of the experimental setup, and then an optimization procedure was used to more precisely estimate the angle and position of the tethered fly within the cylindrical display. Using these measurements, the location of every pixel in the display was mapped to three-dimensional cartesian coordinates relative to the fly position and orientation, and then converted to a pixel-map in spherical coordinates. Additionally, the occlusion by the holder to which the fly is mounted was also transformed into spherical coordinates. The field of view of *Drosophila melanogaster* was estimated based on Andrew Straw's digitized eye map (http://code.astraw.com/drosophila_eye_map/), which was in turn based on E. Buchner's pseudo-pupil mapping reported in his Diplom (*Buchner, 1971*). The composite projection for the estimated positions of every pixel in the display, the position of every ommatidium, and the occlusions due to the holder are shown in *Figure 3—figure supplement 1B*. As the fly was positioned slightly closer to the front of the display from the center of the cylinder, the spacing between pixels is not constant, but averages to 2.5° along the eye equator. The slight offset between the fly's frontal midline and the center of the display, as well as the small offset between the fly eye's equator and the display's horizon capture our best attempt at an accurate alignment. It is important to note that the fly holder occludes large portions of the fly's dorsal and lateral field of view. We believe this alignment to be accurate to within 5°−10°, or the spacing between 1 and 2 ommatidia. Code for this transformation is posted at (https://github.com/reiserlab/LC6downstream; *Morimoto, 2020*; copy archived at swh:1:rev: c76013932b894c3c3abf60d24516d900b47dd9c7).

Looming stimuli were used to map the receptive fields of the LC6G neurons (stimulus type 1, *Supplementary file 2*). Dark discs expanding from ~4.5° to ~18°, at constant velocity of 10°/s, were presented in a grid of 7 × 14 positions, with the centers of the looming stimuli separated by ~9° (*Figure 3*). The grid in *Figure 3A* represents the locations of the looming stimuli centers on the display, while the eye viewing transformation (*Figure 3—figure supplement 1B*) was used to position the stimulus responses more accurately in spherical coordinates from the fly eye perspective (*Figure 3B, C*).

The looming and looming related stimuli used throughout the study are listed in *Supplementary file 2*. Most stimuli progressed with an 'r/v' time course, detailed below. The details of the stimulus types listed in the table (first column) are as follows:

- Small looming RF mapping stimulus: Maximal contrast dark disc, over mid-contrast gray background expanding from ~4.5° to ~18°.
- Dark Looming Disc: Maximal contrast dark disc, over mid-contrast gray background expanding from ~4.5° to ~54°.
- Bright Looming Disc: Maximal contrast bright looming disc stimulus, over mid-contrast gray background expanding from ~4.5° to ~54°.
- Dark Constant Looming Disc: Dark Looming Disc stimulus frames presented with constant speed profile.
- Luminance-matched stimulus:~54° disc darkening with the same temporal profile as Dark Looming Disc stimulus. Intended to isolate the luminance component of the looming stimulus.
- Looming annulus stimulus: Only the edge portion of the Dark Looming Disc stimulus. Intended to isolate the edge-motion component of the looming stimulus.
- Bilateral Looming: Same as Dark Looming Disc but presented at symmetrical positions on both hemi fields of the visual display.
- Small object motion stimulus: Maximal contrast dark ~9° x ~ 9° small square over mid-contrast gray background, sweeping across display.
- Vertical bar motion stimulus: Maximal contrast dark ~9° width, full height vertical bar over mid-contrast gray background, sweeping across display.
- To better match the receptive field of LC6G2 cells, the stimuli were positioned above the equator (18° elevation for looming stimuli, 31.5° elevation for the small object).

For the LC6 neuron RF mapping (*Figure 3—figure supplement 3*), the stimulus was a dark (OFF) flashing square, of size ~9° × ~9°. The square was presented in a pseudorandomized order on a grid spanning ~45° × ~45°, displayed from ~4.5° to ~49.5° from the eye's (anterior) midline in azimuth and from the equator in elevation. The stimuli were presented for 1 s with 2 s intervals.

The stimuli were generated using custom MATLAB (Mathworks, Natick, MA) scripts (https://github.com/mmmorimoto/visual-stimuli). Schematics of the loom-related stimuli are shown in *Figure 4B*. The dark/bright loom stimuli each consisted of a series of 35 disk sizes, with the edge pixel intensity interpolated to approximate a circle on the discrete LED screen. The luminance-matched stimulus was created using the dark looming disk stimulus, spatially scrambling the location of dark pixels of each frame only within the area of the final size of the disk. The looming annulus was created by masking away the inner diameter (leaving the outer 9° diameter) of the dark disc stimulus for every frame. The time series of looming stimuli sizes were presented based on the classic parameterization for looming stimuli assuming a constant velocity of approach. The angular size of the object (θ) increases according to the equation $\theta(t) = 2 \times \tan^{-1}(r/vt)$, where r is the radius of the object, v is its approach speed. Speed of the loom is represented by the ratio of these parameters (r/v, *Figure 4B*, tuning curves) (*Gabbiani et al., 1999*). The experimental protocol consisted of 3 repetitions of each stimulus type presented using a randomized block trial structure. Stimulus epochs were interleaved with at minimum 2 s of blank frame epochs that allowed the GCaMP6m fluorescence to decay back to baseline. Each protocol lasted 15-20 min and subsequently presented three times, resulting in the total experiment time of ~1 hr.

## In vivo two-photon calcium imaging: data analysis

Data were analyzed with software written in MATLAB. Motion correction was performed by cross-correlating each frame to a mean reference image and maximizing the correlation iteratively (using https://bitbucket.org/jastrother/neuron_image_analysis). The fluorescence signal was determined within hand-drawn regions of interest selected to tightly enclose the entire slice of each glomerulus captured within the imaging plane. ΔF/F was calculated as the ratio of $(F - F_0) / F_0$, where F is the instantaneous fluorescence signal and $F_0$ is calculated as the 10th percentile of the fluorescence signal (a robust estimator of baseline levels) within a sliding 300 frame window. These parameters were determined empirically to optimally fit the actual baseline fluorescence. For combining responses of individual flies across animals, we normalized the ΔF/F responses from each individual fly to the 98th percentile of the ΔF/F across all visual stimuli within one experiment. The 98th percentile (a robust estimator of peak levels) was typically near where the curve of the cumulative distribution of all pixel values in an experiment shows saturation. All grouped responses are the mean of the mean response (across repeated stimulus presentations) of all flies in the dataset. For LC6 responses in *Figure 4*, data from 4/6 flies were also used in our previous publication (*Wu et al., 2016*). Error bars for tuning curves and response curves in *Figures 2* and *4* and *Figure 2—figure supplement 1* indicate mean ± SEM. All significance results presented for calcium imaging were determined with the Mann-Whitney U test. Pearson's correlation coefficient (r) was used to assess the degree of correlation.

## EM reconstruction

LC6s and their target neurons were manually reconstructed in a serial section transmission electron microscopy (ssTEM) volume of a full adult female *Drosophila melanogaster* brain (FAFB; *Figure 5A*; *Zheng et al., 2018*) using CATMAID (*Saalfeld et al., 2009*). We followed established guidelines (*Schneider-Mizell et al., 2016*) for tracing skeletons for each neuron and identifying synaptic connections. We found and traced 65 LC6 neurons in the right hemisphere of the FAFB volume but visualized in *Figures 5*, *6* and *7* as the left side. Surprisingly, the LC6/LC9 tract was shifted somewhat medially in this brain, a feature that has also occasionally been observed with light microscopy (http://flycircuit.tw). The morphology of each neuron's dendrites was only approximately reconstructed, favoring completion of the largest branches but not completing the finest dendritic branches. We believe this is adequate for determining the retinotopic location of the center of each neuron's anatomical receptive field, but the boundary of each receptive field might be underestimated. The axon terminals of each LC6 neuron were completely reconstructed and all identifiable synapses on the axons were tagged. The volume demarcated by all LC6 pre-synapses was used to generate a mask identifying the LC6 glomerulus. We then followed the LC6 synapses within the

glomerulus to identify synaptic partners. By comparing the synaptic partners with the morphology of the expected target neurons from light microscopy data (*Figure 1* and *Figure 1—figure supplement 1*), we identified the target neurons which we further reconstructed for the analysis in *Figure 7*. We reconstructed 2 LC6G1 bilateral neurons with cell bodies on the right-hand side and two with cell bodies on the left-hand side. We initially reconstructed two of the ipsilateral LC6G2 neurons as well, but after finding variable receptive field structures we searched extensively and found three more neurons of this same morphology. All target neurons' morphologies (except for the left-hand-side LC6G1 neurons) as well as the connections between the target neurons and LC6 neurons within the right-hand side glomerulus were traced to completion and reviewed by an independent team member.

## Computational analysis of EM reconstructions

All EM analyses and visualizations were carried out in the R language (*R Core Team, 2020*) using open-source packages, mainly the 'natverse' (http://natverse.org/; *Bates et al., 2019*). All reconstructed neurons described in the manuscript are available at https://fafb.catmaid.virtualflybrain.org/, and all data analysis code is available at https://github.com/reiserlab/LC6downstream.

## Retinotopic mapping of the lobula and LC6 neurons

To estimate the retinotopic correspondence for each reconstructed LC6 neuron we mapped the visual field of one eye onto a layer of the lobula. We fit a $2^{nd}$ order curved surface through the dendrites of all LC6 neurons (*Figure 5B,C*). In an ongoing effort we are reconstructing multiple neurons types throughout the medulla to precisely identify the location of every column in the right optic lobe. While this effort is beyond the scope of the work we describe in this manuscript, we used these data to identify two medulla columns that correspond to the center of the eye and a position on the central meridian (the line that partitions the eye between anterior and posterior halves). We expect these points are within one to two columns of these ideal locations. We then identified columnar neurons that connect to LC6 neurons and found one subtype of Tm5 neurons (*Karuppudurai et al., 2014*) that are highly presynaptic to LC6 dendrites in the lobula. We reconstructed Tm5 neurons from the two medulla columns, extending these reference points from the medulla to the surface defined by the LC6 dendrites in the lobula (*Figure 5B,C*). This allowed us to map the curved surface onto a hemisphere. Or more precisely, a spherical lune with [−90˚, +90˚] in longitude and [−10˚, +160˚] in latitude: an approximation of the field of view of one eye (*Buchner, 1971* see eye mapping in *Figure 3—figure supplement 1*, and description above). Consequently, we mapped each LC6 neuron's location (center-of-mass of the dendrite) onto eye coordinates. *Figure 5D* shows the central locations and the boundaries of all LC6 neurons projected onto a flattened field of view of one eye.

## Calculation of the retinotopy index

We developed a quantitative measurement of how much order is preserved by a mapping, that we use in serval different ways to gauge the preservation of retinotopy between the dendrites and axons of LC6 neurons. Used in *Figures 5* and *6*. Suppose we have a point set in a space equipped with a distance metric, and a mapping that maps the point set to a second space with a distance metric (in our intended use case, the first point set are the LC6 dendrite positions in the lobula, and the second is the LC6 axon positions in the glomerulus). Choose a reference point from the set and the rest of the points can be ordered based on their distances to the reference point. The ordered list of points in both spaces can be viewed as a permutation of each other and one can count the number of neighbor swaps (cf. *Bubble sort* in computer science or *Inversion* in combinatorics) required to transform one into the other. The number of swaps quantifies how much disorder the mapping induces. For a given reference point, we define the retinotopy index, RI = $1 - \frac{S}{A} \in [-1, 1]$, where S denotes the number of neighbor swaps and A = (N-1)(N-2)/4 denotes the expected value for the number swaps for N points (including the reference point). The expression for A can be understood as follows: given N-1 ordered points, excluding the reference point, it takes (N-1)(N-2)/2 swaps to completely reverse the order (e.g., 1-2-3-4 → 4-3-2-1). Because of left-right symmetry, that is, for any permutation that is x swaps away from 1-2-3-4, there exist a permutation that is (N-1)(N-2)/2 – x swaps away. Therefore, the expected number of swaps for a random permutation is half of

the maximum. RI = 0 corresponds to a permutation that is (on average) random, while a larger positive value, approaching 1, indicates an order-preserving mapping (negative values correspond to inverted ordering whereby nearby points become more distant than the average distance). The retinotopy index for a given mapping is defined as the average RI over all reference points, denoted by $\bar{RI}$ (but typically refered to more simply as RI), which is the population measure (across cells) that we use in *Figures 5* and *6*.

Note that this retinotopy index is closely related to Kendall distance and Kendall rank correlation coefficient (https://en.wikipedia.org/wiki/Kendall_rank_correlation_coefficient and references therein). The main difference is that we are interested in the amount of order preserved by the mapping (each reference point renders a new permutation), rather than comparing two permutations. So, we take the average value over all reference points and therefore the range and statistical distribution of RI is different from the Kendall coefficient distribution.

RI is by no means a perfect quantifier of retinotopy. Because the ordering of the points is based on a distance metric, the index is sensitive to minor local jittering. For example, 1–2—3 and 1—2–3 are considered different. Moreover, the number of swaps is intrinsically a 1D quantity. A point set in 2D and 3D has many more possible arrangements and therefore, ambiguities. So instead of focusing on the exact value of RI, we typically compare the RI calculated from the data to the RI calculated from a 'control' or a synthetic mapping to gauge its significance.

## Cross-section analysis along LC6 axon tract

The average position of each LC6 neuron's axon was determined at the intersection with the cross-section planes indicated in *Figure 5F*. To get a more accurate position measurement, each LC6 skeleton was re-sampled so as to have nodes placed with approximate uniformity with 400 nm spacing, resulting in ~3 nodes within a 1-μm-thick cross-section. The position of each neuron at each cross-section was taken to be the average position of these nodes. For the visualization in *Figure 5F*, we chose 2 LC6 neurons (one from the anterior group, the other from the posterior group, *Figure 5E*) as references, which defined the positions [0, 0] and [0, 1] in a 2D Cartesian coordinate system at each cross-section. The positions of the rest of the neurons were re-scaled and rotated with respect to the reference neurons (*Figure 5F*).

## Idealized glomerulus projection

*Figures 5E* and *6E* represented a flattened visual field, but in our analysis, these points lay on a curved surface (a spherical lune as described above). The A-P projection in *Figure 6E* was made by simply taking the azimuthal positions of the points. For axes of other angles, we first put all the points on a sphere and rotated around the mid-point of the lobula (corresponding to 73° longitude and 0° latitude in visual field coordinates) by the desired rotation angle, and then took the new azimuthal positions.

## LC6-LC6 connections

We found over 1900 synaptic connections between LC6 neurons in the glomerulus. We tested whether these connections were distributed at random or were more likely to occur between neurons with nearby visual coordinates. For all possible LC6 pairs, we multiplied the distances between dendrite centers in the lobula by the corresponding number of LC6-LC6 synapses in the glomerulus, and then calculated the mean value over all 64 LC6-LC6 pairs for a given LC6 (*Figure 5E*). As a comparison we calculated the weighted mean after randomly shuffling these 64 distances. This shuffling addresses a potential confound in the analysis of biased connectivity originating from the non-uniform density of LC6 dendrite centers. Since the shuffling only scrambles the identity of the neurons, and not their location, the non-homogenous density is also present in the randomized comparisons.

## Anatomical receptive field

For each LC6 neuron, we constructed a 2D Gaussian distribution in the eye coordinate (and example of 2 of these in in *Figure 5E*). The center of each LC6 Gaussian was its dendritic center. Its height was proportional to the synapse count for a given target neuron. The half-widths of all Gaussians were the same and assigned to be proportional to the radius of the dendritic arbors averaged over all 65 LC6 neurons (~30°). The 65 Gaussians were then summed to produce a 2D multi-Gaussian

distribution. This procedure results in an estimate of the feed-forward, anatomical receptive field, based only on neuron morphology and connectivity. To compare this prediction to in vivo imaging data, we have summed the anatomical receptive field estimates from groups of individual neurons to mimic the summed responses from genetically encoded calcium indicator expressed in populations of neurons of the same type (*Figure 7C*, *Figure 7—figure supplement 1*). As these data have been transformed to spherical coordinates in an eye-centered reference frame, they are directly aligned to the receptive fields measured using calcium imaging based on the eye projection (*Figure 3—figure supplement 1*). The composite images are shown in *Figure 7D* and in *Figure 7—figure supplement 2*.

### Glomerulus compartment analysis

We divided the glomerulus in 10 compartments along its long axis, each containing 1/10 of LC6 presynapses (*Figure 6—figure supplement 1B,C*). Since synaptic connections are often polyadic (mostly one-to-many) in fly brains, we used the single presynaptic connectors and ignored the possible numerous postsynaptic connectors for dividing up the glomerulus. For each compartment, we constructed the anatomical receptive field in the same fashion as for the target neurons, except that the height of the Gaussian was proportional to the pre-synapse count in the given compartment.

### Average distance between LC6 neurons and their targets

We defined an average distance between the LC6 neurons and their targets in the glomerulus to explore whether the target neurons preferentially arborize close to the more strongly connected LC6 neurons. We divided the bulk portion of the glomerulus (where most LC6 and target neurons are present) into 80 1-µm-thick slices. We pruned the LC6 axons such that within each slice, an LC6's nodes defined a unique position. For a target neuron, we collected all its nodes that fell in that slice (*Figure 7—figure supplement 2*). The distance of this target neuron to the selected LC6 neurons (see *Figure 7—figure supplement 2* legend and discussion in the main text for the selection criterion) was the mean distance between all LC6-target node pairs.

### EM analysis: hemibrain data

We accessed the hemibrain dataset (release 'hemibrain:v1.1') (*Scheffer et al., 2020*) with neuPrintExplorer (https://neuprint.janelia.org/) and neuprint-python (https://connectome-neuprint.github.io/neuprint-python/docs/). We only included reconstructions with status 'Traced' and synapses associated with such reconstructions in our analyses. We further excluded synapses within the optic lobe (regions 'LO(R)', 'ME(R)' and 'LP(R)'); this largely limits the analyzed synapses of LC cells to connections within their respective glomeruli. LC6 and other LC types were identified based on existing hemibrain annotations (e.g. type='LC6'). For LC6, we visually confirmed that the 61 named LC6 reconstructions had the expected morphology (see *Figure 8—figure supplement 2B*). For comparison to light microscopy and computational searches, hemibrain skeletons were aligned to the JRC2018U template brain (*Bogovic et al., 2018*). The illustrations in *Figure 8—figure supplement 1A* and *Figure 8—figure supplement 2B* were assembled using VVD viewer.

To find candidate LC6G reconstructions in the hemibrain dataset, we examined the anatomy of LC6 pre- and postsynaptic partners and also used computational search tools (*Clements and Goina, 2020*; *Otsuna et al., 2018*) to look for cells similar to light microscopy images of LC6G neurons. The reconstructions we identified as likely instances of LC6G cell types are listed in *Supplementary file 4* and illustrated in *Figure 8—figure supplement 1A*. *Supplementary file 4* also includes some reconstructions that are similar to LC6G cells but that we considered different cell types, as further described below. We also confirmed that our control cell type (LC26G1) is an LC26 target, as predicted from light microscopy (832 synapses from LC26, two synapses from LC6 and 22 from LC16).

Because the hemibrain is largely limited to one brain hemisphere, LC6G1 and similar bilateral cells are not fully contained in the reconstructed volume. We therefore used synaptic connectivity in addition to cell morphology to identify likely LC6G1 projections from the contralateral brain (LC6G1L in *Figure 8—figure supplement 1* and *Supplementary file 4*): The seven hemibrain bodies that we annotated as LC6G1L cells are both pre- and postsynaptic to LC6 neurons with a total of at least 30 synaptic connections in each direction, while other cells of similar morphology (presumably the left counterparts of PVLP008_b, PVLP008_c and PVLP0007) have at most ten combined synapses

with LC6 cells. Ipsilateral LC6G1 cells differed from similar cells in shape and position: All but one of the ipsilateral PVLP008_b and PVLP008_c cells show an additional ipsilateral branch that is absent from LC6G1 cells. PVLP007 cells (annotated as 'LC16G1' in *Supplementary file 4*) are similar to LC6G1 but connect the LC16 instead of the LC6 glomeruli.

All LC6G cell types are small groups of similar cells and, as types, were readily matched to groups of hemibrain reconstructions. In some cases, details of anatomy and synaptic connectivity suggest that these groups could be further subdivided: For example, two hemibrain cell types (PVLP133_a, 5 cells and PVLP133_b, 3 cells) match the morphology of LC6G2 LM and EM images and were both classified as LC6G2. Both receive input from LC16 and LC6. By contrast, the most similar related hemibrain cell type (PVLP133_c) shows only very limited overlap with the LC16 and LC6 glomeruli and has almost no synapses with LC6 and LC16. Nine cells of two hemibrain types (PVLP58 and PVLP59) that are anatomically similar to LC6G5 cells were both tentatively identified as LC6G5, though there are substantial differences in the LC6 and LC16 connections within this group. Connectivity differences that may point to subtypes were also observed within other groups (e.g. LC16 synapses of the two PVLP006/LC6 G3 cells appear different). Because of this, while we are confident that our annotations identify the LC6G cells in the hemibrain in general, it is possible that some individual reconstructions annotated as LC6G cells are not part of the driver line patterns and may represent distinct cell types. The LC6G3 driver lines also appear to label a few cells with input to the LC9 glomerulus (similar to hemibrain cell types PVLP004 and PVLP005) and the LC6G5 driver lines appear to include some cells with a contralateral projection similar to PVLP052 (a minor LC16 target) or one of several related cell types.

## Acknowledgements

We thank the Janelia Fly Light Project Team for help with imaging driver lines and processing of FISH samples. We thank Jasmine Le for training and assistance with the functional connectivity experiments and Hideo Otsuna and John Bogovic for help with registration of hemibrain skeletons. We thank members of the Bock lab, especially Scott Lauritzen, for supporting our early EM reconstruction efforts and Gregory Jefferis for introducing us to his tools for computational neuroanatomy. We thank Janelia's Connectome Annotators: Padideh Ghorbani, N Aidan Smith, Marisa Dreher, Miriam Flynn, Connor Laughland, Henrique Ludwig, Alex Thomson, Bruck Gezahegn, supervised by Ruchi Parekh, for their dedicated reconstruction of the neuronal circuits featured in this study. Development and administration of the FAFB tracing environment and analysis tools were funded in part by National Institutes of Health BRAIN Initiative grant 1RF1MH120679-01 to Davi Bock and Greg Jefferis, with software development effort and administrative support provided by Tom Kazimiers (Kazmos GmbH) and Eric Perlman (Yikes LLC). This work was made possible in part by software funded by the NIH: Fluorender: An Imaging Tool for Visualization and Analysis of Confocal Data as Applied to Zebrafish Research, R01-GM098151-01. We are also grateful to members of the Reiser Lab, especially Eyal Gruntman, Frank Loesche, and Kit Longden, for comments on the manuscript. This project was supported by HHMI.

## Additional information

### Funding

| Funder | Author |
| --- | --- |
| Howard Hughes Medical Institute | Mai M Morimoto<br>Aljoscha Nern<br>Arthur Zhao<br>Edward M Rogers<br>Allan M Wong<br>Mathew D Isaacson<br>Davi D Bock<br>Gerald M Rubin<br>Michael B Reiser |

The funders had no role in study design, data collection and interpretation, or the decision to submit the work for publication.

## Author contributions
Mai M Morimoto, Conceptualization, Software, Formal analysis, Investigation, Methodology, Writing - original draft, Writing - review and editing; Aljoscha Nern, Conceptualization, Resources, Software, Formal analysis, Investigation, Writing - original draft, Writing - review and editing; Arthur Zhao, Software, Formal analysis, Writing - original draft, Writing - review and editing; Edward M Rogers, Resources; Allan M Wong, Mathew D Isaacson, Software, Methodology; Davi D Bock, Resources, Funding acquisition, Methodology, Writing - review and editing; Gerald M Rubin, Funding acquisition, Writing - review and editing; Michael B Reiser, Conceptualization, Supervision, Funding acquisition, Methodology, Writing - original draft, Writing - review and editing

## Author ORCIDs
Mai M Morimoto (iD) https://orcid.org/0000-0002-9654-3960
Aljoscha Nern (iD) https://orcid.org/0000-0002-3822-489X
Arthur Zhao (iD) https://orcid.org/0000-0003-2869-4393
Edward M Rogers (iD) https://orcid.org/0000-0002-8275-1830
Allan M Wong (iD) http://orcid.org/0000-0002-8492-2162
Mathew D Isaacson (iD) https://orcid.org/0000-0001-8797-0090
Davi D Bock (iD) http://orcid.org/0000-0002-8218-7926
Gerald M Rubin (iD) http://orcid.org/0000-0001-8762-8703
Michael B Reiser (iD) https://orcid.org/0000-0002-4108-4517

## Decision letter and Author response
Decision letter https://doi.org/10.7554/eLife.57685.sa1
Author response https://doi.org/10.7554/eLife.57685.sa2

# Additional files

## Supplementary files
• Supplementary file 1. Detailed summary of genotypes used throughout the manuscript.

• Supplementary file 2. Detailed summary of visual stimuli.

• Supplementary file 3. Connectivity matrix of LC6 and target neurons from EM data.

• Supplementary file 4. Hemibrain reconstructions of LC6G cell types and related cells (three tables). Table 1. provides details of cells identified as likely matches of LC6G cell types in the hemibrain dataset as well as some related cells considered in the annotation process. The table includes the hemibrain identifiers ('hemibrain_bodyId'), hemibrain cell type names ('hemibrain_type'), LC6G annotations ('new_type'), synaptic connections with LC6, LC16 and LC26 in the central brain and some notes related to the LC6G annotations. Table 2. lists all LC6 targets in the hemibrain (excluding other LC6 cells) that receive more than 100 combined central brain synapses from all LC6 neurons. Table 3. lists LC6 inputs in the hemibrain (excluding other LC6 cells) that provide more than 30 combined central brain synapses to all LC6 neurons.

• Transparent reporting form

## Data availability
All data generated or analyzed during this study are included in the manuscript and supporting files. Source data files have been provided for Figures 3, 5 and 6 along with analysis code (https://github.com/reiserlab/LC6downstream) copy archived at https://archive.softwareheritage.org/swh:1:rev:c76013932b894c3c3abf60d24516d900b47dd9c7/. All reconstructed neurons described in the manuscript are available at https://fafb.catmaid.virtualflybrain.org/.

The following previously published datasets were used:

| Author(s) | Year | Dataset title | Dataset URL | Database and Identifier |
|---|---|---|---|---|
| Lauritzen Z | 2018 | FAFB | https://temca2data.org/ | Temca2, FAFB |
| Scheffer L | 2020 | hemibrain:v1.1 | https://neuprint.janelia.org/?dataset=hemibrain:v1.1 | hemibrain:v1.1, hemibrain:v1.1 |

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
