## [Decision Letter]

**Acceptance summary:**

This paper is of very significant interest. Using light and EM microscopy, it analyzes the pathway in (LC neurons), and downstream of optic glomeruli. These are conceptually important structures that process visual 'features' and emphasize these features as compared to retinotopy. You showed that although fine retinotopy is to a large extent lost as LC neurons cover overlapping domains in the glomeruli, neurons downstream of LC6 do exhibit spatial selectivity, suggesting that processing in higher centers is responsible. This is impressive work and opens the door to investigating the functioning of optic glomeruli.

**Decision letter after peer review:**

Thank you for submitting your article "Connectivity establishes spatial readout of visual looming in a glomerulus lacking retinotopy" for consideration by *eLife*. Your article has been reviewed by Ronald Calabrese as the Senior Editor, a Reviewing Editor, and three reviewers. The following individual involved in review of your submission has agreed to reveal his identity: Markus Meister (Reviewer #3).

The reviewers have extensively discussed the reviews with one another and the Reviewing Editor has drafted this decision to help you prepare a revised submission.

The reviewers found the topic of great interest and the work to be done very carefully and in depth, including light and EM microscopy that is very impressive. However, they could not be convinced of the major claim of the paper, that retinotopy is lost in the glomeruli.

– Optic glomeruli are conceptually very important structures that process visual 'features' and emphasize these features as compared to retinotopy. But is the latter lost (see below) although LC neurons cover the map quite precisely? Maybe it is your choice of a looming glomerulus for which retinotopy might not be that important? Therefore, even if retinotopy was partially lost in this glomerulus, is it a general mechanism for all glomeruli?

– It appears that none of the reviewers is convinced by the fact that the (weak) spatial organization does not account for the (also rather weak) spatial biases. The projection and synaptic sites of LC6 are already non-uniform, and the degree of spatial selectivity that is seen in LC6Gs is likely still present in the circuitry.

– On the other hand, your work identifies novel candidate neurons downstream of LC6, demonstrates functional connectivity, shows some evidence for spatial selectivity within these neurons, and starts to look at the microcircuitry they are in.

Therefore, we suggest to reframe your data with less of a bias to demonstrate a loss of retinotopy and use these data to demonstrate what the paper is doing best: identify downstream pathway of the optic glomeruli and explore neural processing.

Reviewer #1:

The manuscript "Connectivity establishes spatial readout of visual looming in a glomerulus lacking retinotopy" provides an identification and characterization of neurons downstream of the LC6 glomerulus. This is an interesting paper, with includes a wealth of data, and sophisticated analysis of physiology and connectivity of LC6 and its downstream neuron candidates.

While the paper should in principle be interesting to a large neuroscience audience, there is currently a mismatch between the results that I was expecting from reading abstract and text, and the results shown in the data. I still think that the data itself are exciting and that this a very thorough study, but need to be more accurately reflected in the writing.

Essential revisions:

I am not fully convinced that the LC6 glomerulus fully loses retinotopy. The data in Figure 1 convincingly show that LC6 terminals widely innervate the LC6 glomerulus. However, LC6G neurons also widely respond to looming stimuli across the visual field, and just show preferences (enhanced responses) in different regions. How can the authors exclude the possibility that there still is coarse retinotopy in this glomerulus, that would well explain the spatial selectivity of the LC6G neurons.

It is feasible that an uneven distribution of LC6 innervation is just not visible from a light microscopic level of analysis as shown? Indeed, the authors are coming back to saying that retinotoy of LC6 projections is not visible from light microscopy (e.g. – Results section), but how does this look like in the projections of LC6 axons in the EM data, which has much higher resolution? This question should be addressable from the wealth of EM data analyzed for this manuscript. How does coverage of the LC6 glomerulus look like for individual neurons, especially those that cover the most lateral glomerulus compartment (#1), corresponding to the anterior visual field.

It is not clear to me why the authors instead look for 'higher level' features in the connectivity such as LC6 to LC6 synapses.

I am basically wondering if there could still be rough topography of terminals in the glomeruli, that would be fully sufficient to explain the spatial biases in the responses of the LC6G neurons. If this was true, I still think that this would be a great story, but it's important to clarify this point.

In Figure 3, I cannot fully follow your description of the data, and I was expecting spatially much more restricted responses from reading the text. For example, LC6G1 and LC6G2 neurons do not really respond to a "restricted portion of the visual field". Especially LC6G1 neurons respond to (almost) anywhere within the visual field, as the time traces show. They indeed respond *stronger* to specific regions in the visual field, and thus display some degree of spatial selectivity. What mostly convinces me that there is some spatial selectivity (at least in the amplitude of the response) is the result that several neurons from one type have very similar responses. This could be explicitly mentioned, whereas some other statements should be toned down.

In the description of these results, it is also confusing what "when imaged as a population" (subsection “Neurons downstream of LC6 exhibit stereotyped spatial receptive fields”) means. I cannot find these population imaging data. I also don't see how individual neurons are spatially more selective individually than the population. To draw this conclusion from selectively looking at 60% peak contour lines (which is still a very high response) is misleading.

In general, I agree with the statement that these results support a (very coarse) spatially selective read-out, but statements should be toned down. In the Abstract, I'd for example suggest to change "respond to looming in different portions of the visual field" to "more strongly respond to looming in different portions.…", or similar.

Similar to the comment above, the description of Figure 5 seems to overstate some of the findings. For example, I agree that compartment #1 is biased towards the posterior visual field. But compartment 10 isn't much biased towards the posterior visual field, but instead pretty evenly covers the visual field with peak responses to the center. Remaining compartments (esp. #5-9) don't show much of a bias.

Later in the paper, the authors describe the results in Figure 5 as "weak evidence for a continuous spatial organization ("retinotopy") (Discussion section). Why can this weak topographical organization not account for the (also rather weak) spatial biases? That seems like a very plausible explanation, and wouldn't make this a less interesting paper.

By the way, I don't think one should talk about "retinotopy" here or anywhere else, but rather about spatial selectivity, or spatial organization, as done in most parts of the paper.

After reconstructing the LC6G1 and 2 neurons, can't the authors first look at the anatomical coverage of these neurons (or e.g. their synapses / postsynaptic sites) across the LC6 glomerulus? As evident from the comments above, I am not convinced that there isn't coarse topography within the glomerulus. The example cell shown in Figure 6A mostly covers the lateral compartments, which together with coarse retinotopy in LC6 neurons, could then also explain the spatially biased receptive field (Figure 6C).

Subsection “Connectivity-based readout of visual information in the LC6 glomerulus corroborates features of the spatially selective responses of target neurons”: I do not see "substantial agreement" between the EM-estimated RFs and measured RFs in Figure 6D.

Subsection “Connectivity-based readout of visual information in the LC6 glomerulus corroborates features of the spatially selective responses of target neurons”: I do not find sufficient support to agree with the statement that the data "clearly establish that spatially selective readout by optic glomerulus neurons is accomplished using biased connectivity within a structure in the central brain where the neuronal arbors themselves are not retinotopically arranged".

Despite all the criticism above, I fully agree with the summary of the paper as given at the beginning of the Discussion section. The authors identify candidate downstream neurons, demonstrate functional connectivity, show that these neurons have similar / more selective looming responses as compared to LC6, and there is some evidence from spatial selectivity.

If this spatial selectivity might or might not be inherited from a coarse topographic organization of LC6 neurons that was missed so far and is only visible in EM data, is still unclear. If the authors could address this point more clearly, and not overinterpret any results, I'd find this a suitable study for publication in *eLife* (and no matter what the result is).

Reviewer #2:

This study addresses a new question in fly visual neuroscience: does spatial vision emerge from the axon terminals of visual projection neurons forming the so-called 'optic glomeruli' of the protocerebrum? The authors provide functional evidence for the identity of downstream neurons, and then show that the EM measured pattern of synaptic connections corresponds with functional receptive field measurements of the downstream cells. That the spatial receptive fields exist at all defies the notion that retinotopy is indeed lost. The strengths of the paper surely lie in the exquisite EM reconstruction and mapping of synaptic connections to the visual field. The comparisons between anatomical and functional relationships are less compelling. I would have the following key criticisms.

1) It would be very helpful to see measured LC6 RFs, and comparison of functional and EM-anatomical RFs.

2) LC6-LC6 synapses are vastly overrepresented in synaptic connectivity by these cells in the glomerulus. Yet, there is no assessment of their functional impact on LC6 visual tuning. Nor are such putative interactions summarized in the model circuit diagram in Figure 7. I hate to suggest experiments, but this is a key unexpected finding of the paper, that LC6-LC6 axo-axonic connectivity is so strong (has that been shown for other LCs? Not to my knowledge…but maybe it hasn't been tested with Den-Syt?). A simple experiment would be to co-express Shi(ts) and GCaMP6f in LC6 and compare intact and feedback blocked visually evoked responses. I actually wonder if the authors don't have such data in hand, since they developed similarly complex reagents for the functional connectivity analysis.

3) Blocking LC6 ought also to have profound impacts on LCG'X' responses unless the G neurons receive significant additional input from elsewhere, which seems likely given the anatomical spread of the dendrites of all but LC6G1. As it is written, the manuscript tends to convey the notion that LC6G neurons receive all or most of their inputs from LC6 but that is not tested (e.g. Introduction).

4) Figure 5: There is clearly an overrepresentation of LC6 anatomical RFs near midline (panel D), so wouldn't you therefore expect bias in the 'nearest neighbor' analysis in panel E?

5) In Subsection “Connectivity-based readout of visual information in the LC6 glomerulus corroborates features of the spatially selective responses of target neurons” the authors highlight the congruence of EM-estimated and measured RFs by LC6G neurons. However, the functional RFs in G1 and G2 are huge. The singular 70% cutoff contour is not representative of the size or shape of these RFs. How much would the interpretation of similarity differ if a 50% contour is used? Or a 90% contour? Per my point, from visual inspection, the maxima of the EM-anatomical RFs do not match the functional ones at all for either G1 or G2, with the peaks instead separated by roughly 45-degrees (G1) or more (G2). Similarly, there are prominent minima in the EM-estimated RF, particularly for G1 at [0,100] that 'corresponds' with prominent measured responses. Why not represent both anatomical and functional RFs with the nice blue heat maps as in 6C, which would allow for a more objective comparison of the EM-estimated and measured RFs? I realize that some interpolation would be required, but so what? And then provide a 2-D spatial cross-correlation of values? Or similar objective comparison to bootstrapped distributions?

Reviewer #3:

This article is about neurons and circuits in a late stage of the *Drosophila* visual system. The lobula is a brain area with a retinotopic organization of neurons, meaning cells that are near each other have receptive fields at nearby locations in space. From the lobula some neurons project to the visual glomeruli: These are clumps of neuropil within which the retinotopic organization seems to have been lost. Here the authors ask what happens to the spatially-specific signals that enter the glomerulus: are they hopelessly intermixed because of the commingling of axonal arbors. The report identifies output neurons from the glomerulus and inspects their visual responses using optical recording and their anatomical structure from EM reconstruction. On that background the paper makes two main claims:

1) Glomerulus output neurons have spatially restricted receptive fields even though the inputs to the glomerulus show no retinotopy. The results can be explained based on selective synaptic connectivity assessed from EM reconstructions.

2) Circuits within the glomerulus establish spatial readout of visual features and contralateral suppression.

Essential revisions:

1) The evidence for (1) is weak. The summary is in Figure 6. Only 2 output neuron types were inspected, G1 and G2. The physiology says that their RFs are almost completely disjoint (red-cyan in Figure 6D). EM analysis says that one RF should be fully contained inside the other (blue-blue in Figure 6D). This is not "a substantial level of agreement" (Subsection “Connectivity-based readout of visual information in the LC6 glomerulus corroborates features of the spatially selective responses of target neurons”). The Figure 6 caption promises additional comparisons between anatomical and functional RFs but I could not find them. It is difficult to claim that selective synaptic connections explain neural responses based on this one case with poor agreement.

2) The suggestions about circuits postsynaptic to the glomerulus are interesting but incomplete. In the present report they appear more as an afterthought (Figure 7). Only two of the postsynaptic neuron types (G1, G2) are in this circuit, whereas the authors identified three additional ones (G3, G4, G5). Presumably those could also be traced through the EM stacks. One would want to know where they go before speculating about overall function of these circuits.

Regarding the logic of the presentation, what exactly is the authors' null hypothesis? That synapses are slaves to a retinotopic arrangement of the presynaptic axonal arbors? But *Drosophila* neuroscience teaches us the opposite: There exist countless combinations of cell surface molecules that organize specific synaptic connections within a dense neuropil where different signals are intermingled. Of course, that's true in vertebrates as well. The emergence of direction-selective responses within retinal circuits would be impossible based on retinotopy alone. The same holds for all sorts of other neural computations. See the recent Sanes and Zipursky review. On that background the findings reported here are unsurprising. But I am not sure they further illuminate the issues of selective connectivity.

---

## [Author Response]

The reviewers found the topic of great interest and the work to be done very carefully and in depth, including light and EM microscopy that is very impressive. However, they could not be convinced of the major claim of the paper, that retinotopy is lost in the glomeruli.– Optic glomeruli are conceptually very important structures that process visual 'features' and emphasize these features as compared to retinotopy. But is the latter lost (see below) although LC neurons cover the map quite precisely? Maybe it is your choice of a looming glomerulus for which retinotopy might not be that important? Therefore, even if retinotopy was partially lost in this glomerulus, is it a general mechanism for all glomeruli?– It appears that none of the reviewers is convinced by the fact that the (weak) spatial organization does not account for the (also rather weak) spatial biases. The projection and synaptic sites of LC6 are already non-uniform, and the degree of spatial selectivity that is seen in LC6Gs is likely still present in the circuitry.– On the other hand, your work identifies novel candidate neurons downstream of LC6, demonstrates functional connectivity, shows some evidence for spatial selectivity within these neurons, and starts to look at the microcircuitry they are in.Therefore, we suggest to reframe your data with less of a bias to demonstrate a loss of retinotopy and use these data to demonstrate what the paper is doing best: identify downstream pathway of the optic glomeruli and explore neural processing.

We thank the editor for this very helpful summary. We are pleased to learn that the reviewers found our work to be well conducted and our topic to be of great conceptual importance. The primary concerns raised center around whether retinotopy is truly ‘lost’ in the glomerulus. We now understand that in the original submission we had not adequately answered this question. From our perspective, the glomerulus plainly does not show any of the obviously retinotopic characteristics of the fly optic lobe, but we had not fully embraced the considerable challenge of examining whatever residual retinotopic organization might be present in the glomerulus. In this thoroughly updated manuscript, we have now explored this organization in great detail. To summarize our findings, we now have much stronger evidence that some spatial organization is present in the glomerulus, but simply exploiting this organization cannot account for the selective readout by the target neurons. We have also substantially modified the paper with a more balanced presentation, in line with the suggestion to reframe the results. We have revised the Title, Abstract, Introduction and the text throughout, so that we no longer claim a ‘lack of retinotopy’ as one of our main finding.

The question of whether retinotopy is more or less important in a looming feature pathway is indeed a very interesting question, which we speculate about in subsection “Spatial detection of visual looming”. However, we have no reason to believe that LC6s form an atypical glomerulus. All glomeruli (with the exception of LC10) show a similar intermingling at the gross scale that is observable with light microscopy (as described in our previous work on LC types (Wu et al., 2016)) and as we now describe, all LC axon terminals are interconnected with other LCs of the same type (new Figure 8—figure supplement 2). Evaluating the presence of visual-spatial organization in other optic glomeruli will require the type of data we produced for the LC6 pathway, which is currently not available for any other glomeruli. However, the analysis methods we’ve introduced in this paper should address these questions and provide a framework for a broader inquiry into spatial processing in the fly central brain, once these data are established for the other visual pathways.

In the revised manuscript we have endeavored to formalize the measurement of retinotopy within a neuropil. We are aware of rigorous work to quantify e.g. the tiling of neurons in the retina, but have not seen a method for measuring retinotopy, which is usually only described in qualitative terms. We now introduce a ‘Retinotopy Index’ that quantifies the ‘orderliness’ of a mapping that transforms a set of points relative to an initial ordering. This quantitative tool allows us to examine different aspects of whether LC6 neurons that are nearby in the input structure (the lobula), are nearby in the output structure (the glomerulus)—used in updated Figure 5, Figure 6. We have also verified major connectomic results in our data set by analyzing the latest ‘hemibrain’ connectome data, providing ‘N=2’ confirmation for the claims about connectivity within the glomerulus and about the bilateral inputs (Figure 8 and new Figure 8—figure supplement 1, Figure 8—figure supplement 2). We believe that this significant expansion of our EM analysis (Figure 5 and Figure 6) provides additional rigor and clarity to our current work, and now makes this manuscript more relevant to the wider visual neuroscience community.

We hope this revision fully addresses the concerns of the reviewers on the problem of retinotopy and bring other aspects of the paper more into focus.

Reviewer #1:The manuscript "Connectivity establishes spatial readout of visual looming in a glomerulus lacking retinotopy" provides an identification and characterization of neurons downstream of the LC6 glomerulus. This is an interesting paper, with includes a wealth of data, and sophisticated analysis of physiology and connectivity of LC6 and its downstream neuron candidates.While the paper should in principle be interesting to a large neuroscience audience, there is currently a mismatch between the results that I was expecting from reading abstract and text, and the results shown in the data. I still think that the data itself are exciting and that this a very thorough study, but need to be more accurately reflected in the writing.

We thank the reviewer for these incredibly thoughtful comments. We are grateful that the reviewer appreciates the topic and quality of our work and understand all of the concerns raised about how we discuss and present our results. We have significantly updated the manuscript and have addressed all points raised below (we have numbered the points for convenience).

Essential revisions:1) I am not fully convinced that the LC6 glomerulus fully loses retinotopy. The data in Figure 1 convincingly show that LC6 terminals widely innervate the LC6 glomerulus. However, LC6G neurons also widely respond to looming stimuli across the visual field, and just show preferences (enhanced responses) in different regions. How can the authors exclude the possibility that there still is coarse retinotopy in this glomerulus, that would well explain the spatial selectivity of the LC6G neurons.It is feasible that an uneven distribution of LC6 innervation is just not visible from a light microscopic level of analysis as shown? Indeed, the authors are coming back to saying that retinotoy of LC6 projections is not visible from light microscopy (e.g. Results section), but how does this look like in the projections of LC6 axons in the EM data, which has much higher resolution? This question should be addressable from the wealth of EM data analyzed for this manuscript. How does coverage of the LC6 glomerulus look like for individual neurons, especially those that cover the most lateral glomerulus compartment (#1), corresponding to the anterior visual field.It is not clear to me why the authors instead look for 'higher level' features in the connectivity such as LC6 to LC6 synapses.I am basically wondering if there could still be rough topography of terminals in the glomeruli, that would be fully sufficient to explain the spatial biases in the responses of the LC6G neurons. If this was true, I still think that this would be a great story, but it's important to clarify this point.

These comments are very pointed and helpful! In the initial submission we had taken a fairly narrow view on the question of retinotopy—in our view, of course retinotopy was lost in the glomerulus since the axons (and their synapses) of neurons conveying visual signals from all parts of the eye were intermingled. This motivated us to look at the ‘higher level,’ rather indirect, evidence for structure in the LC6-LC6 synapses. Because of the helpful comments of the reviewers, we now realize that we had skipped several steps in our data analysis and presentation, and the EM data provides a much better opportunity to re-examine these questions. We have now updated Figure 5 and include a new Figure 6 that show this more direct analysis requested here, detailing the distribution of processes and synapses in the glomerulus. We’ve developed a (to our knowledge) novel quantification of retinotopy and applied this retinotopy index to the glomerulus at different scales (axon skeleton median position, median position of the pre-synapses, and the distance between pre-synapses of different neurons within compartments). While we cannot rule out some very intricate fine-scale organization, what we find is that in most ways of looking at this, the glomerulus has only modest, but significantly non-random visual-spatial structure. The strongest evidence for retinotopy is observed along the long axis of the glomerulus which roughly corresponds to the dorso-ventral axis of the visual field (Figure 6C-D). This coarse organization is consistent with the non-random LC6-LC6 connectivity, a result which now makes a supporting point alongside these new analyses and has been placed as new Figure 6—figure supplement 1.

Based on this new analysis, we conclude that there is indeed some retinotopic organization in the LC6 glomerulus, but it is far from what one sees in the visual system. This organization can be used by downstream cells to extract spatial information, however, this is unlikely to be the sole mechanism by which the spatial bias in LC6G RFs arise, since the downstream cells sample the glomerulus in a largely homogeneous manner (Figure 7—figure supplement 3). This is not a simple question to resolve definitively (akin to proving a negative), but we believe this new analysis based on comparing the proximity of LC6G1 and LC6G2 neurons to the region of the eye with the largest connectivity difference (Figure 7—figure supplement 2), clearly shows that the target neurons must do some ‘fine scale’ recognition of their connected neuron partners.

2) In Figure 3, I cannot fully follow your description of the data, and I was expecting spatially much more restricted responses from reading the text. For example, LC6G1 and LC6G2 neurons do not really respond to a "restricted portion of the visual field". Especially LC6G1 neurons respond to (almost) anywhere within the visual field, as the time traces show. They indeed respond *stronger* to specific regions in the visual field, and thus display some degree of spatial selectivity. What mostly convinces me that there is some spatial selectivity (at least in the amplitude of the response) is the result that several neurons from one type have very similar responses. This could be explicitly mentioned, whereas some other statements should be toned down.In the description of these results, it is also confusing what "when imaged as a population" (subsection “Neurons downstream of LC6 exhibit stereotyped spatial receptive fields”) means. I cannot find these population imaging data. I also don't see how individual neurons are spatially more selective individually than the population. To draw this conclusion from selectively looking at 60% peak contour lines (which is still a very high response) is misleading.In general, I agree with the statement that these results support a (very coarse) spatially selective read-out, but statements should be toned down. In the Abstract, I'd for example suggest to change "respond to looming in different portions of the visual field" to "more strongly respond to looming in different portions.…", or similar.

We apologize for this confusion. Our initial discussion was unclear about two points: we did not explain well that individual measurements were from a population of neurons in a single fly, not from individual cells, and we confused the discussion by summarizing the LC6G1 and LC6G2 responses together (when they are quite different, as the reviewer points out). This section has been updated to clarify these points. We have also implemented all of the specific suggestions made here. We have updated the text and figure legend, and also updated the figure to clarify what is plotted in panels B and C. We now explicitly mention the consistency in the spatial responses of the individual types. We apologize for the confusion about the selectivity of individual cells versus the population—we never measured individual cell responses, and so we are simply making the inference that single neurons could exhibit sharper spatial selectivity than what we did measure from groups of neurons. Hopefully this is now clarified.

In addition, the Abstract and Introduction have also been updated to remove claims about ‘spatially restrictive’ or ‘spatially selective’ and replaced with ‘spatially biased’.

3) Similar to the comment above, the description of Figure 5 seems to overstate some of the findings. For example, I agree that compartment #1 is biased towards the posterior visual field. But compartment 10 isn't much biased towards the posterior visual field, but instead pretty evenly covers the visual field with peak responses to the center. Remaining compartments (esp. #5-9) don't show much of a bias.

We agree with this comment and now describe the estimated RFs from each compartment more accurately. The finding of an anterior-dorsal cluster of LC6 neurons that occupy compartments 1-3 (Figure 6C,D) has further clarified our interpretation. This compartment-wise analysis is also now much less central since it has been supplanted by the new results in Figure 5 and Figure 6 and has been moved to Figure 6—figure supplement 1.

4) Later in the paper, the authors describe the results in Figure 5 as "weak evidence for a continuous spatial organization ("retinotopy") (Discussion section). Why can this weak topographical organization not account for the (also rather weak) spatial biases? That seems like a very plausible explanation, and wouldn't make this a less interesting paper.By the way, I don't think one should talk about "retinotopy" here or anywhere else, but rather about spatial selectivity, or spatial organization, as done in most parts of the paper.

This is a very fair question, and one we had not dealt with adequately in the original submission. In the updated Figure 5 and Figure 6 we have a much-improved analysis of the retinotopy in the glomerulus and directly address whether this organization is likely to account for the spatial biases in the downstream neurons. In Figure 7—figure supplement 2, Figure 7—figure supplement 3 we now explore in more detail how the coarse spatial organization of the LC6 axons in the glomerulus does not appear to be sufficient to account for the spatial biases in downstream RFs.

We agree with the cautious note about not using the term retinotopy when spatial organization makes more sense, and we have been careful to do so in the revised manuscript. However, we find that ‘spatial organization’ is in some places more confusing, since it could describe the spatial organization of axons within the glomerulus with no connection to the spatial organization of the visual system, which is why we now use retinotopy specifically where our Retinotopy Index allows us to directly compare the organization of the glomerulus back to the lobula.

5) After reconstructing the LC6G1 and 2 neurons, can't the authors first look at the anatomical coverage of these neurons (or e.g. their synapses / postsynaptic sites) across the LC6 glomerulus? As evident from the comments above, I am not convinced that there isn't coarse topography within the glomerulus. The example cell shown in Figure 6A mostly covers the lateral compartments, which together with coarse retinotopy in LC6 neurons, could then also explain the spatially biased receptive field (Figure 6C).

We have now done exactly this analysis, and it makes up Figure 7—figure supplement 2 and Figure 7—figure supplement 3. The reviewer is correct that there is some spatial organization within the glomerulus (most prominent is that along the longitudinal axis of the glomerulus we find a rough correspondence to the dorso-ventral axis of the visual field [Figure 6C,D]). We analyzed the coverage of the LC6G neurons and find that they sample the glomerulus is a largely similar manner (Figure 7—figure supplement 3), and with a synapse density that only varies by about ±50% (per unit of cable length). We further examined whether the spatial biases throughout the glomerulus can explain the different receptive fields of different LC6G neurons (Figure 7—figure supplement 2) and find that they do not appear to. Perhaps the simplest way to see that gross organization of the glomerulus is insufficient to account for the selectivity of the downstream neurons is that the strongest organization we find is along the D-V axis, while the connectivity that best distinguishes LC6G1 and LC6G2 is along the A-P axis.

6) Subsection “Connectivity-based readout of visual information in the LC6 glomerulus corroborates features of the spatially selective responses of target neurons”: I do not see "substantial agreement" between the EM-estimated RFs and measured RFs in Figure 6D.

We have changed “agreement” to “overlap”, and the comparison between the EM and the functional measurements is now focused around the differences between these cell types, which is best exemplified by the ‘δ’ of the anatomical receptive fields (Figure 7—figure supplement 2).

7) Subsection “Connectivity-based readout of visual information in the LC6 glomerulus corroborates features of the spatially selective responses of target neurons”: I do not find sufficient support to agree with the statement that the data "clearly establish that spatially selective readout by optic glomerulus neurons is accomplished using biased connectivity within a structure in the central brain where the neuronal arbors themselves are not retinotopically arranged".

We have updated and simplified this sentence: Taken together this connectomic analysis of the LC6 glomerulus suggests that spatially selective readout by optic glomerulus neurons is accomplished using fine-scale, biased connectivity (Discussion section).

Despite all the criticism above, I fully agree with the summary of the paper as given at the beginning of the Discussion section. The authors identify candidate downstream neurons, demonstrate functional connectivity, show that these neurons have similar / more selective looming responses as compared to LC6, and there is some evidence from spatial selectivity.If this spatial selectivity might or might not be inherited from a coarse topographic organization of LC6 neurons that was missed so far and is only visible in EM data, is still unclear. If the authors could address this point more clearly, and not overinterpret any results, I'd find this a suitable study for publication in eLife (and no matter what the result is).

We thank the reviewer again for these very thoughtful and helpful comments. In addressing these comments, we have dramatically revised the manuscript, and it is now much improved!

Reviewer #2:This study addresses a new question in fly visual neuroscience: does spatial vision emerge from the axon terminals of visual projection neurons forming the so-called 'optic glomeruli' of the protocerebrum? The authors provide functional evidence for the identity of downstream neurons, and then show that the EM measured pattern of synaptic connections corresponds with functional receptive field measurements of the downstream cells. That the spatial receptive fields exist at all defies the notion that retinotopy is indeed lost. The strengths of the paper surely lie in the exquisite EM reconstruction and mapping of synaptic connections to the visual field. The comparisons between anatomical and functional relationships are less compelling. I would have the following key criticisms.

We are grateful for this positive assessment of our work. Based on the reviewer’s constructive criticisms, we have revised the manuscript as detailed below.

1) It would be very helpful to see measured LC6 RFs, and comparison of functional and EM-anatomical RFs.

We have now included Figure 3—figure supplement 3, that shows LC6 RFs as measured by GCaMP responses from putative boutons in the glomerulus. These measurements are imperfect—due to the dense packing of boutons from many LC6 axons in the glomerulus, it is unlikely that most pixels correspond to single LC6 neurons (this is why these data were not included in the original manuscript), but nevertheless these experiments provide a rough estimate of RF size (for this mapping stimulus of not more than ~20 degrees) that is consistent, but perhaps on the low side, of the anatomical estimates.

2) LC6-LC6 synapses are vastly overrepresented in synaptic connectivity by these cells in the glomerulus. Yet, there is no assessment of their functional impact on LC6 visual tuning. Nor are such putative interactions summarized in the model circuit diagram in Figure 7. I hate to suggest experiments, but this is a key unexpected finding of the paper, that LC6-LC6 axo-axonic connectivity is so strong (has that been shown for other LCs? Not to my knowledge…but maybe it hasn't been tested with Den-Syt?). A simple experiment would be to co-express Shi(ts) and GCaMP6f in LC6 and compare intact and feedback blocked visually evoked responses. I actually wonder if the authors don't have such data in hand, since they developed similarly complex reagents for the functional connectivity analysis.

We agree that the intricate connectivity between LC6 axons in the glomerulus is quite striking and is only unexpected in the sense that such connections have not been carefully examined before. From our ongoing, as yet unpublished, connectomics reconstruction of other visual projection neurons, and from the new hemibrain data set (Scheffer et al., 2020), we now know that connections (in the optic lobe and in the central brain) between neighboring projection neurons are very common, and that LC6 is in no way remarkable in comparison to other LCs.

As to whether these connections are ‘vastly overrepresented’ – it is unclear what is the most meaningful comparison. Due to our reconstruction process, we attempted to annotate all synapses between LC6 neurons and the target cells, but we do not have complete counts of all inputs/outputs of all LC6 neurons. For the smaller subset of LC6 neurons that have all input and output synapses tagged, we find between 3-5% of the output synapses are on to other LC6 neurons (approx. 40 out of 1000 total), while nearly 30% of the input synapses come from other LC6 neurons (approx. 30 out of 100 total). The LC6 inputs to the axon terminal seem quite significant, however (so far as we know) there is no strong evidence for the impact of this pre-synaptic excitation. The majority of other inputs are presumed to be inhibitory, and therefore we have no way of directly comparing the relative contributions of these inputs. Furthermore, since this 30% of inputs comes from, on average 29 other LC6 neurons, very few of which would be activated by a typical looming visual stimulus, it is likely to be a weak input at any given point in time. As our data are incomplete, we have also examined the recent hemibrain data set (Discussion section: “assessing major results in light of new fly central brain connectomic data” and Figure 8 —figure supplement 2) and confirm that most LC neurons feature prominent axo-axonic connections in their respective glomeruli, and so this is unlikely to be a special computational adaptation for the detection of looming stimuli. While we lack any definitive data, our best guess is that the impact of these LC6-LC6 connections is subtle, much more like a modest somewhat-spatial activity averaging, than a major determinant of LC6 synaptic release. We have augmented the discussion to more thoroughly cover this topic (subsection “Spatial detection of visual looming”).

We do not have data for the clever experiments the reviewer has suggested, and we did not pursue these experiments (squelching LC6 synaptic release while measuring visual stimulus evoked calcium responses in the LC6 axon terminals) since we expect these effects would be quite hard to see. This expectation is informed by our experience working on the motion detection circuitry, where our group, as well as several others, have carried out the exact experiment suggested, while investigating similar, excitatory, connections between the direction-selective T4 cells (dendro-dendritic have been described in the published EM studies, but we have also found substantial axo-axonic connections as well). No consistent effects were found in the T4 responses to visual motion. Understanding the role of these excitatory pre-synaptic connections is a very interesting question, that we regard as being outside of the scope of this paper, potentially requiring more sensitive measurement and manipulation methods.

3) Blocking LC6 ought also to have profound impacts on LCG'X' responses unless the G neurons receive significant additional input from elsewhere, which seems likely given the anatomical spread of the dendrites of all but LC6G1. As it is written, the manuscript tends to convey the notion that LC6G neurons receive all or most of their inputs from LC6 but that is not tested (e.g. Introduction).

The reviewer is absolutely correct that neurons other than LC6 are likely to provide inputs to the LC6G neurons. We cannot completely answer this question based on our targeted reconstruction, but we have now included a survey of LC6 targets in the hemibrain dataset, and address whether they receive inputs from other LC neurons in subsection “Assessing major results in light of new fly central brain connectomic data” and in Figure 8 —figure supplement 1 and the tables in Supplementary file 4. By providing hemibrain identifiers for LC6G reconstructions (Supplementary file 4—table 1), we also make it easy for readers to further explore LC6G connectivity in this public dataset. We agree with the reviewer’s suggestion that blocking LC6 neurons is expected to dramatically reduce looming responses of most LC6G neurons (the situation may be more complicated for cells that also receive substantial input from LC16, another looming responsive LC ). We have not carried out these experiments, but we have added a discussion of this (Discussion section) and suggest that these would be important follow-ups in understanding the integration of signals between different LC neuron types. We have now elaborated more on the potential limitations of our anatomical RF reconstruction methods (subsection “Limitations of comparisons between the functional and anatomical receptive fields of LC6 downstream cells“), and mention the fact that network effects, such as input from other LC types or other glomerulus neurons are not taken into account (Discussion section). We hope this clarifies that LC6 is not the sole input to the LC6 downstream cells we investigate in this paper.

4) Figure 5: There is clearly an overrepresentation of LC6 anatomical RFs near midline (panel D), so wouldn't you therefore expect bias in the 'nearest neighbor' analysis in panel E?

This is a very reasonable concern, but the comparison to shuffled data handles this potential confound. Since the shuffling only scrambles the identity of the neurons, and not their location, the non-homogenous density is also present in the randomized comparisons. We have added an explanation of this to the Materials and methods section for this analysis.

5) In Subsection “Connectivity-based readout of visual information in the LC6 glomerulus corroborates features of the spatially selective responses of target neurons” the authors highlight the congruence of EM-estimated and measured RFs by LC6G neurons. However, the functional RFs in G1 and G2 are huge. The singular 70% cutoff contour is not representative of the size or shape of these RFs. How much would the interpretation of similarity differ if a 50% contour is used? Or a 90% contour? Per my point, from visual inspection, the maxima of the EM-anatomical RFs do not match the functional ones at all for either G1 or G2, with the peaks instead separated by roughly 45-degrees (G1) or more (G2). Similarly, there are prominent minima in the EM-estimated RF, particularly for G1 at [0,100] that 'corresponds' with prominent measured responses. Why not represent both anatomical and functional RFs with the nice blue heat maps as in 6C, which would allow for a more objective comparison of the EM-estimated and measured RFs? I realize that some interpolation would be required, but so what? And then provide a 2-D spatial cross-correlation of values? Or similar objective comparison to bootstrapped distributions?

We have provided the blue heat-mapped versions of the functional RFs, now as Figure 3—figure supplement 2, to facilitate further comparisons. In addition, we now included Figure 7—figure supplement 2 in which we overlay the functional RFs on the difference between the EM estimated RFs (G1-G2 or G2-G1). This allows for visual comparison of functional and EM estimated RFs with more emphasis on the difference between the two cell types. We have also included a thorough discussion on the many limitations of comparing the anatomical and functional RFs (subsection “Limitations of comparisons between the functional and anatomical receptive fields of LC6 downstream cells”). We believe the comparison has quite a lot of value, and while we do not find a precise match, many features do agree, and LC6G neurons are found to support a spatially biased readout.

Reviewer #3:This article is about neurons and circuits in a late stage of the *Drosophila* visual system. The lobula is a brain area with a retinotopic organization of neurons, meaning cells that are near each other have receptive fields at nearby locations in space. From the lobula some neurons project to the visual glomeruli: These are clumps of neuropil within which the retinotopic organization seems to have been lost. Here the authors ask what happens to the spatially-specific signals that enter the glomerulus: are they hopelessly intermixed because of the commingling of axonal arbors. The report identifies output neurons from the glomerulus and inspects their visual responses using optical recording and their anatomical structure from EM reconstruction. On that background the paper makes two main claims:1) Glomerulus output neurons have spatially restricted receptive fields even though the inputs to the glomerulus show no retinotopy. The results can be explained based on selective synaptic connectivity assessed from EM reconstructions.2) Circuits within the glomerulus establish spatial readout of visual features and contralateral suppression.

We appreciate the clarity with which the reviewer summarizes our claims and have addressed the issues presented below. We’d like to point out that our paper, in addition to making these claims, is also an exploration of a previously un-investigated part of the fly brain, and so we ask the reviewer for some credit for the exploratory and descriptive work that preceded either of these claims.

Essential revisions:1) The evidence for (1) is weak. The summary is in Figure 6. Only 2 output neuron types were inspected, G1 and G2. The physiology says that their RFs are almost completely disjoint (red-cyan in Figure 6D). EM analysis says that one RF should be fully contained inside the other (blue-blue in Figure 6D). This is not "a substantial level of agreement" (Subsection “Connectivity-based readout of visual information in the LC6 glomerulus corroborates features of the spatially selective responses of target neurons”). The Figure 6 caption promises additional comparisons between anatomical and functional RFs but I could not find them. It is difficult to claim that selective synaptic connections explain neural responses based on this one case with poor agreement.

In response to this comment and those of the other reviewers, we have updated the data presentation, and ‘toned down’ many of the claims, especially about the level of agreement between the EM-predicted RFs and functional data. We have added Figure 7—figure supplement 2 and Figure 7—figure supplement 3 that help the two-way comparison between the functional and EM estimated RFs for the differences between the cells. These difference plots remove the confounding peaks in the response that had more to do with non-uniform density of RF centers for the LC6 neurons. We have also added subsection “Limitations of comparisons between the functional and anatomical receptive fields of LC6 downstream cells” that details the many challenges that limit the level of agreement that could be expected between these very different methods for estimated receptive fields. We agree with the conclusion of the reviewer – on the basis of these results (Figure 7) we cannot claim that the direct synaptic connectivity between LC6s and the LC6Gs can explain the neural responses, but we do believe they clearly demonstrate that spatially biased readout of the projection neurons is carried out in the glomerulus, and have toned down our claims accordingly.

2) The suggestions about circuits postsynaptic to the glomerulus are interesting but incomplete. In the present report they appear more as an afterthought (Figure 7). Only two of the postsynaptic neuron types (G1, G2) are in this circuit, whereas the authors identified three additional ones (G3, G4, G5). Presumably those could also be traced through the EM stacks. One would want to know where they go before speculating about overall function of these circuits.

We thank the review for agreeing that the analysis of the interconnectivity between the glomeruli is interesting, and we of course, also acknowledge that it is incomplete. While the proposal may appear at the end of the paper, it is certainly not an afterthought. The reconstructed circuit shown here already represents more than 1.5 person-years of manual reconstruction. We don’t claim a complete circuit analysis here, but we believe the evidence already supports the proposal of a pathway for bilateral suppression, based on a direct connection, that is almost certainly inhibitory, between the LC6 glomeruli. To bolster this proposal, we have now explored the new hemibrain data set to survey a much larger set of connected neurons in the glomerulus. This survey is summarized in subsection “Assessing major results in light of new fly central brain connectomic data” and two new supplementary figures (and Figure 8—figure supplement 1 and Figure 8—figure supplement 2) with additional data in Supplementary file 4. This larger set of neurons confirms the original proposal, that the direct connection between the glomeruli (via LC6G1 neurons) is the shortest pathway that could mediate inhibition in response to contralateral looming stimuli (for which we provided functional evidence, in now Figure 8). We also confirm our finding that contralateral LC6G1 cells are an input to LC6G2. The hemibrain data further indicate that LC6G1 neurons are unique in the high number (both absolute and relative to their total pre- and postsynapses) of input and output synapses to LC6 cells. Thus, while more complex pathways between the left and right LC6 glomeruli likely exist, LC6G1 cells, the central component of the circuit we detail in Figure 8, almost certainly have a key role in the communication between the two LC6 glomeruli.

Regarding the logic of the presentation, what exactly is the authors' null hypothesis? That synapses are slaves to a retinotopic arrangement of the presynaptic axonal arbors? But *Drosophila* neuroscience teaches us the opposite: There exist countless combinations of cell surface molecules that organize specific synaptic connections within a dense neuropil where different signals are intermingled. Of course, that's true in vertebrates as well. The emergence of direction-selective responses within retinal circuits would be impossible based on retinotopy alone. The same holds for all sorts of other neural computations. See the recent Sanes and Zipursky review. On that background the findings reported here are unsurprising. But I am not sure they further illuminate the issues of selective connectivity.

The null hypothesis (based on light microscopy) is that the LC6 glomerulus lacks any structure or organization and consequently, spatial vision within the glomerulus would not be possible. This was not meant as a “straw man,” but as a serious expectation based on prior work. Even if there is specific connectivity within the glomerulus, it is not necessarily the case that summing up the direct synaptic inputs to the glomerulus should predict the receptive fields of the output neurons. Instead, substantial connectivity within the glomerulus could establish the selectivity of the output neurons (this point is now raised as one of several limitations of our analysis in subsection “Limitations of comparisons between the functional and anatomical receptive fields of LC6 downstream cells”). Importantly, while the prior reports about a lack of intraglomerular structure of LC6 (and other LC) terminals at the light microscopy level does not necessarily mean that retinotopic information is discarded, it also does not imply that it is preserved or, if so, reveal to what degree it is preserved. This is an empirical question that requires physiological or ultrastructural data, such as those presented here, to answer an important question for understanding the flow of spatial visual information in the fly brain.

We appreciate this pointed critique from the reviewer, but we submit that the circuit we’ve examined here is different in several important ways from the previously described examples. We do not believe the literature contains many descriptions of selective wiring within a group of cells, of the same type, that reside in a not-very-structured neuropil. Given the number of LC6 neurons, and the presence of multiple target cell types, each potentially representing a different biased spatial selectivity, the molecular logic of this wiring specificity could well be more complicated than the previously described examples (probably the most impressive being the exquisite organization of the inputs to the T4/T5 neurons in the fly visual system – which is highly ordered, and in which the selectivity is evidenced by different types of neurons providing inputs at different spatial locations). This is discussed in the Discussion section. We agree with the reviewer that there is no fundamental paradox about specific connectivity within a seemingly disorganized structure (given the diversity of potentially available cell recognition molecules). But this “in principle” answer is quite a long way from understanding the underlying developmental mechanisms. To quote the cited review “Despite considerable progress in understanding early steps in neural development, we are a long way from understanding how the neural circuits are assembled that underlie perception, cognition, emotion, action, and much more.” Examples of circuits with distinct organizational features, such as the one described in our work, may well be of interest for understanding principles of circuit assembly (in addition to circuit function, which is our focus here).